# Shallow VAEs with RealNVP Prior Can Perform as Well as Deep Hierarchical VAEs

## Abstract

Using powerful posterior distributions is a popular technique in variational inference. However, recent works showed that the aggregated posterior may fail to match unit Gaussian prior, even with expressive posteriors, thus learning the prior becomes an alternative way to improve the variational lower-bound. We show that using learned RealNVP prior and just one latent variable in VAE, we can achieve test NLL comparable to very deep state-of-the-art hierarchical VAE, outperforming many previous works with complex hierarchical VAE architectures. We hypothesize that, when coupled with Gaussian posteriors, the learned prior can encourage appropriate posterior overlapping, which is likely to improve reconstruction loss and lower-bound, supported by our experimental results. We demonstrate that, with learned RealNVP prior, $\beta$-VAE can have better rate-distortion curve than using fixed Gaussian prior.

## 1 Introduction

Variational auto-encoder (VAE) (Kingma & Welling, 2014; Rezende et al., 2014) is a powerful deep generative model. The use of *amortized variational inference* makes VAE scalable to deep neural networks and large amount of data. Variational inference demands the intractable true posterior to be approximated by a tractable distribution. The original VAE used factorized Gaussian for both the prior and the variational posterior (Kingma & Welling, 2014; Rezende et al., 2014). Since then, lots of more expressive variational posteriors have been proposed (Tran et al., 2016; Rezende & Mohamed, 2015; Salimans et al., 2015; Nalisnick et al., 2016; Kingma et al., 2016; Mescheder et al., 2017; van den Berg et al., 2018). However, recent work suggested that even with powerful posteriors, VAE may still fail to match *aggregated posterior* to unit Gaussian prior (Rosca et al., 2018), indicating there is still a gap between the approximated and the true posterior.

To improve the variational lower-bound, one alternative way to using powerful posterior distributions is to learn the prior, an idea initially suggested by Hoffman & Johnson (2016). Later on, Huang et al. (2017) applied RealNVP (Dinh et al., 2017) to learn the prior. Tomczak & Welling (2018) proved the optimal prior is the *aggregated posterior*, which they approximate by assembling a mixture of the posteriors with a set of learned pseudo-inputs. Bauer & Mnih (2019) constructed a rich prior by multiplying a simple prior with a learned acceptance function. Takahashi et al. (2019) introduced the *kernel density trick* to estimate the KL divergence in ELBO, for their implicit prior.

Despite the achievements of these previous works on posteriors and priors, the state-of-the-art VAE models with continuous latent variables all rely on deep hierarchical latent variables[1], although some of them might have used complicated posteriors/priors as components in their architectures. Most latent variables in such deep hierarchical VAEs have no clear semantic meanings, just a technique for reaching good lower-bounds. This is in sharp contrast with GANs (Goodfellow et al., 2014; Arjovsky et al., 2017) and flow-based models (Ho et al., 2018), where most of them can reach state-of-the-art results with only one latent variable, with clear semantic meanings. We thus raise and answer a question: **with the help of learned priors, can shallow VAEs achieve performance comparable or better than deep hierarchical VAEs?** This question is important because a shallow

---

[1]The term "hierarchical latent variables" refers to multiple layers of latent variables, formulated as $p(\mathbf{x}) = \mathbb{E}_{p(\mathbf{z}_1)}\mathbb{E}_{p(\mathbf{z}_2|\mathbf{z}_1)}\cdots\mathbb{E}_{p(\mathbf{z}_K|\mathbf{z}_{K-1},\dots,\mathbf{z}_1)}[p(\mathbf{x}|\mathbf{z}_1,\dots,\mathbf{z}_K)]$; while "one latent variable" refers to just one $\mathbf{z}$ in standard VAEs. Both $\mathbf{z}_k$ and $\mathbf{z}$ are multi-dimensional tensors. Also, "deep" refers to many hierarchical latent variables, while "shallow" refers to few latent variables.

VAE would be much more promising to scale to more complicated datasets than deep hierarchical VAEs. To answer this question, we conduct comprehensive experiments on several datasets with learned RealNVP priors and just one latent variable, which even shows advantage over some deep hierarchical VAEs with powerful posteriors. We also propose a hypothesis on why learning the prior can lead to such great improvement in model performance, supported by our experimental results. In summary, our contributions are:

- We conduct comprehensive experiments on four binarized datasets with four different network architectures. Our results show that VAE with RealNVP prior consistently outperforms standard VAE and RealNVP posterior.

- We are the first to show that using learned RealNVP prior with just one latent variable in VAE, it is possible to achieve test negative log-likelihoods (NLLs) comparable to very deep state-of-the-art hierarchical VAE on these four datasets, outperforming many previous works using complex hierarchical VAE equipped with rich priors/posteriors.

- We hypothesize that, when coupled with Gaussian posteriors, the learned prior can encourage appropriate posterior overlapping, which is likely to improve reconstruction loss and lower-bound, supported by our experimental results.

- We demonstrate that, with learned RealNVP prior, $\beta$-VAE can have better rate-distortion curve (Alemi et al., 2018) than using fixed Gaussian prior.

## 2 PRELIMINARIES

### 2.1 VARIATIONAL AUTO-ENCODER

Variational auto-encoder (VAE) (Kingma & Welling, 2014; Rezende et al., 2014) is a deep probabilistic model. It uses a latent variable $\mathbf{z}$ with prior $p_\lambda(\mathbf{z})$, and a conditional distribution $p_\theta(\mathbf{x}|\mathbf{z})$, to model the observed variable $\mathbf{x}$. $p_\theta(\mathbf{x})$ is defined as $\int_{\mathcal{Z}} p_\theta(\mathbf{x}|\mathbf{z}) \, p_\lambda(\mathbf{z}) \, \mathrm{d}\mathbf{z}$, where $p_\theta(\mathbf{x}|\mathbf{z})$ is derived by a neural network with parameter $\theta$. $\log p_\theta(\mathbf{x})$ is bounded below by evidence lower-bound (ELBO):

$$\log p_\theta(\mathbf{x}) \geq \mathcal{L}(\mathbf{x}; \lambda, \theta, \phi) = \mathbb{E}_{q_\phi(\mathbf{z}|\mathbf{x})} \left[ \log p_\theta(\mathbf{x}|\mathbf{z}) + \log p_\lambda(\mathbf{z}) - \log q_\phi(\mathbf{z}|\mathbf{x}) \right] \tag{1}$$

$$= \mathbb{E}_{q_\phi(\mathbf{z}|\mathbf{x})} \left[ \log p_\theta(\mathbf{x}|\mathbf{z}) \right] - D_{\mathrm{KL}}(q_\phi(\mathbf{z}|\mathbf{x}) \| p_\lambda(\mathbf{z})) \tag{2}$$

where $q_\phi(\mathbf{z}|\mathbf{x})$ is the variational posterior to approximate $p_\theta(\mathbf{z}|\mathbf{x})$, derived by a neural network with parameter $\phi$. In Eq. (2), the first term is the *reconstruction loss* of $\mathbf{x}$, and the second term is the Kullback Leibler (KL) divergence between $q_\phi(\mathbf{z}|\mathbf{x})$ and $p_\lambda(\mathbf{z})$. Optimizing $q_\phi(\mathbf{z}|\mathbf{x})$ and $p_\theta(\mathbf{x}|\mathbf{z})$ *w.r.t.* empirical distribution $p^\star(\mathbf{x})$ can be achieved by maximizing the expected ELBO *w.r.t.* $p^\star(\mathbf{x})$:

$$\mathcal{L}(\lambda, \theta, \phi) = \mathbb{E}_{p^\star(\mathbf{x})}[\mathcal{L}(\mathbf{x}; \lambda, \theta, \phi)] = \mathbb{E}_{p^\star(\mathbf{x})} \mathbb{E}_{q_\phi(\mathbf{z}|\mathbf{x})} \left[ \log p_\theta(\mathbf{x}|\mathbf{z}) + \log p_\lambda(\mathbf{z}) - \log q_\phi(\mathbf{z}|\mathbf{x}) \right] \tag{3}$$

A hyper-parameter $\beta$ can be added to the training objective, in order to control the trade-off between reconstruction loss and KL divergence, known as $\beta$-VAE (Higgins et al., 2017; Alemi et al., 2018):

$$\mathcal{L}_\beta(\lambda, \theta, \phi) = \mathbb{E}_{p^\star(\mathbf{x})} \mathbb{E}_{q_\phi(\mathbf{z}|\mathbf{x})} \left[ \log p_\theta(\mathbf{x}|\mathbf{z}) + \beta \left( \log p_\lambda(\mathbf{z}) - \log q_\phi(\mathbf{z}|\mathbf{x}) \right) \right] \tag{4}$$

The *aggregated posterior* $q_\phi(\mathbf{z}) = \int_{\mathcal{X}} q_\phi(\mathbf{z}|\mathbf{x}) \, p^\star(\mathbf{x}) \, \mathrm{d}\mathbf{x}$, should be equal to $p_\lambda(\mathbf{z})$, if VAE is perfectly trained (Hoffman & Johnson, 2016). However, even with powerful posteriors, the *aggregated posterior* may still not match a unit Gaussian prior (Rosca et al., 2018). To investigate this problem, Hoffman & Johnson (2016) suggested to decompose Eq. (3) as:

$$\mathcal{L}(\lambda, \theta, \phi) = \underbrace{\mathbb{E}_{p^\star(\mathbf{x})} \mathbb{E}_{q_\phi(\mathbf{z}|\mathbf{x})} \left[ \log p_\theta(\mathbf{x}|\mathbf{z}) \right]}_{\text{①}} - \underbrace{D_{\mathrm{KL}}(q_\phi(\mathbf{z}) \| p_\lambda(\mathbf{z}))}_{\text{②}} - \underbrace{\mathbb{I}_\phi[Z; X]}_{\text{③}} \tag{5}$$

where $\mathbb{I}_\phi[Z; X] = \iint q_\phi(\mathbf{z}, \mathbf{x}) \log \frac{q_\phi(\mathbf{z}, \mathbf{x})}{q_\phi(\mathbf{z}) \, p^\star(\mathbf{x})} \mathrm{d}\mathbf{z} \, \mathrm{d}\mathbf{x}$ is the *mutual information*. Since $p_\lambda(\mathbf{z})$ is only in ②, ELBO can be further enlarged if $p_\lambda(\mathbf{z})$ is trained to match $q_\phi(\mathbf{z})$.

### 2.2 REALNVP

RealNVP (Dinh et al., 2017) is a deep probabilistic model. We denote its observed variable by $\mathbf{z}$ and latent variable by $\mathbf{w}$, with marginal distribution $p_\lambda(\mathbf{z})$ and prior $p_\xi(\mathbf{w})$. Unlike VAE, RealNVP

relates $\mathbf{z}$ and $\mathbf{w}$ by an invertible mapping $\mathbf{w} = f_\lambda(\mathbf{z})$ instead of a conditional distribution, as follows:

$$p_\lambda(\mathbf{z}) = p_\xi(\mathbf{w}) \left| \det\left(\frac{\partial f_\lambda(\mathbf{z})}{\partial \mathbf{z}}\right) \right|, \quad \mathbf{z} = f_\lambda^{-1}(\mathbf{w}) \tag{6}$$

where $\det(\partial f_\lambda(\mathbf{z})/\partial \mathbf{z})$ is the Jacobian determinant of $f_\lambda$. In RealNVP, $f_\lambda$ is composed of $K$ invertible mappings, where $f_\lambda(\mathbf{z}) = (f_K \circ \cdots \circ f_1)(\mathbf{z})$, and each $f_k$ is invertible. $f_k$ must be carefully designed to ensure that the determinant can be computed efficiently. The original paper of RealNVP introduced the *affine coupling layer* as $f_k$. Kingma & Dhariwal (2018) further introduced *actnorm* and *invertible 1x1 convolution*. Details can be found in their papers.

## 3    LEARNING THE PRIOR WITH REALNVP

It is straightforward to obtain a rich prior $p_\lambda(\mathbf{z})$ from a simple (*i.e.*, with constant parameters) one with RealNVP. Denote the simple prior as $p_\xi(\mathbf{w})$, while the RealNVP mapping as $\mathbf{w} = f_\lambda(\mathbf{z})$. We then obtain Eq. (6) as our prior $p_\lambda(\mathbf{z})$. Substitute Eq. (6) into (3), we get to the training objective:

$$\mathcal{L}(\lambda, \theta, \phi) = \mathbb{E}_{p^\star(\mathbf{x})} \mathbb{E}_{q_\phi(\mathbf{z}|\mathbf{x})} \left[ \log p_\theta(\mathbf{x}|\mathbf{z}) + \log p_\xi(f_\lambda(\mathbf{z})) + \log \left| \det\left(\frac{\partial f_\lambda(\mathbf{z})}{\partial \mathbf{z}}\right) \right| - \log q_\phi(\mathbf{z}|\mathbf{x}) \right] \tag{7}$$

We mainly use *joint training* (Tomczak & Welling, 2018; Bauer & Mnih, 2019) (where $p_\lambda(\mathbf{z})$ is jointly trained along with $q_\phi(\mathbf{z}|\mathbf{x})$ and $p_\theta(\mathbf{x}|\mathbf{z})$ by directly maximizing Eq. (7)), but we also consider other two strategies, see Appendices B.3 and B.9 for discussions.

In our initial experiments, training VAEs with deep RealNVP priors is unstable. The HVAEs (see Appendix B.13) with RealNVP priors are even more unstable, where the training procedure can barely finish. However, we tackle this problem by the following two techniques: (1) using gradient clip, as the original RealNVP (Dinh et al., 2017) does; and (2) clipping the std of Gaussian posterior $q_\phi(\mathbf{z}|\mathbf{x})$ of VAE (or $q_\phi(\mathbf{z}_2|\mathbf{x})$ and $q_\phi(\mathbf{z}_1|\mathbf{x})$ of HVAE) by a minimum value of $e^{-11} \approx 1.67 \times 10^{-5}$.

## 4    APPROPRIATE POSTERIOR OVERLAPPING WITH LEARNED PRIOR

As we shall see in Section 5.4, RealNVP prior can result in a test negative log-likelihood (NLL) substantially better than RealNVP posterior, with the same flow depth. Also, using both RealNVP posterior and prior shows no significant advantage over using RealNVP prior only, although the total flow depth of the former variant is twice as large as the latter one. To investigate why VAE with RealNVP prior can lead to substantially better model performance than the other two variants, we take the particular architecture of our models into consideration (*i.e.*, Bernoulli $p_\theta(\mathbf{x}|\mathbf{z})$, Gaussian $q_\phi(\mathbf{z}|\mathbf{x})$, and unit Gaussian or flow prior $p_\lambda(\mathbf{z})$), and start with the following proposition:

**Proposition 1.** *Given a finite number of discrete training data, i.e., $p^\star(\mathbf{x}) = \frac{1}{N} \sum_{i=1}^{N} \delta(\mathbf{x} - \mathbf{x}^{(i)})$, if $p_\theta(\mathbf{x}|\mathbf{z}) = \text{Bernoulli}(\boldsymbol{\mu}_\theta(\mathbf{z}))$, where the Bernoulli mean $\boldsymbol{\mu}_\theta(\mathbf{z})$ is produced by the decoder, and $0 < \mu_\theta^k(\mathbf{z}) < 1$ for each of its $k$-th dimensional output, then the optimal decoder $\boldsymbol{\mu}_\theta(\mathbf{z})$ is:*

$$\boldsymbol{\mu}_\theta(\mathbf{z}) = \sum_i w_i(\mathbf{z}) \mathbf{x}^{(i)}, \quad \text{where } w_i(\mathbf{z}) = \frac{q_\phi(\mathbf{z}|\mathbf{x}^{(i)})}{\sum_j q_\phi(\mathbf{z}|\mathbf{x}^{(j)})} \text{ and } \sum_i w_i(\mathbf{z}) = 1 \tag{8}$$

*Proof.* See Appendix A.    $\square$

Proposition 1 suggests that if $q_\phi(\mathbf{z}|\mathbf{x})$ for different $\mathbf{x}$ overlap, then even at the center of $q_\phi(\mathbf{z}|\mathbf{x}^{(i)})$ of one training point $\mathbf{x}^{(i)}$, the optimal decoder will be an *average* of both $\mathbf{x}^{(i)}$ and other training points $\mathbf{x}^{(j)}$, weighted by $w_i(\mathbf{z})$ and $w_j(\mathbf{z})$. Rezende & Viola (2018) have shown that weighted average like this is likely to cause poor *reconstruction loss* (① in Eq. (5)) and "blurry reconstruction". Besides poor reconstructions, when $\mathbf{x}$ is high-dimensional, the reconstruction loss would typically dominate ELBO, suggesting that good reconstruction loss is necessary for obtaining good ELBO.

To enlarge ①, it is a crucial goal to appropriately reduce the overlapping of $q_\phi(\mathbf{z}|\mathbf{x})$ for different $\mathbf{x}$. For Gaussian posterior, this seems to involve shrinking the standard deviations (std) of most $q_\phi(\mathbf{z}|\mathbf{x})$,

since when the mean of every $q_\phi(\mathbf{z}|\mathbf{x})$ is determined, smaller stds for the majority of $q_\phi(\mathbf{z}|\mathbf{x})$ would effectively result in less overlapping.

When the prior $p_\lambda(\mathbf{z})$ is fixed (such as unit Gaussian), however, smaller stds for $q_\phi(\mathbf{z}|\mathbf{x})$ are likely to induce "holes" on the aggregated posterior, causing $q_\phi(\mathbf{z})$ to be more dissimilar with the prior, enlarging ② in Eq. (5). Also, less overlapping among the Gaussian posteriors $q_\phi(\mathbf{z}|\mathbf{x})$ would correspond to higher *mutual information* (③ in Eq. (5)) (Hoffman & Johnson, 2016; Mathieu et al., 2019). As a result, with a fixed prior, there should exist a trade-off between ① and ② + ③. In fact, such a trade-off has already been discussed in the context of $\beta$-VAE (Alemi et al., 2018).

It should be a desirable property that the overlapping among $q_\phi(\mathbf{z}|\mathbf{x})$ would be large for similar $\mathbf{x}$, and conversely, small for dissimilar $\mathbf{x}$. We shall use the term "appropriate overlapping" to denote such property. We believe that the trade-off between the reconstruction loss (①) and the mutual information (③) should be one important ingredient for reaching the "appropriate overlapping", as implied by their respective meanings. However, the effect of ② is not so clear. We suspect that, in many cases, this term may exhibit over-regularization, resulting in too large overlapping, and further unsatisfactory reconstruction loss. In this paper, we choose to use learned RealNVP prior to reduce the effect from ②, such that the trade-off would occur mainly between ① and ③. We hope this will bring us closer to the "appropriate overlapping", than using the fixed unit Gaussian prior.

In conclusion, when $p_\theta(\mathbf{x}|\mathbf{z})$ is Bernoulli, we hypothesize that learning the prior can encourage "appropriate overlapping" among Gaussian posteriors, which is vital for good reconstructions and NLLs. Rezende & Viola (2018) has proved when $p_\theta(\mathbf{x}|\mathbf{z}) = \mathcal{N}(\boldsymbol{\mu}_\theta(\mathbf{z}), \sigma^2 \mathbf{I})$ with fixed constant $\sigma$, the optimal decoder is also $\boldsymbol{\mu}_\theta(\mathbf{z}) = \sum_i w_i(\mathbf{z}) \mathbf{x}^{(i)}$, thus our analysis also holds in such situation.

## 5 EXPERIMENTS

### 5.1 SETUP

**Datasets**    We use four datasets in our experiments: statically and dynamically binarized MNIST (Larochelle & Murray, 2011; Salakhutdinov & Murray, 2008) (denoted as *StaticMNIST* and *MNIST* in our paper, respectively), FashionMNIST (Xiao et al., 2017) and Omniglot (Lake et al., 2015). Details of these datasets can be found in Appendix B.1.

**Models**    We perform systematically controlled experiments, using the following VAE variants: (1) **DenseVAE**, with dense layers; (2) **ConvVAE**, with convolutional layers; (3) **ResnetVAE**, with ResNet layers (Zagoruyko & Komodakis, 2016); and (4) **PixelVAE** (Gulrajani et al., 2017), with several PixelCNN layers on top of the ResnetVAE decoder. For RealNVP priors and posteriors, we use $K$ blocks of invertible mappings ($K$ is called *flow depth* hereafter), while each block contains an *invertible dense*, a dense *coupling layer*, and an *actnorm* (Dinh et al., 2017; Kingma & Dhariwal, 2018). The dimensionality of $\mathbf{z}$ are 40 for StaticMNIST and MNIST, while 64 for FashionMNIST and Omniglot. More details are in Appendix B.2. We also conduct experiments with **ResnetHVAE** and **PixelHVAE** (Tomczak & Welling, 2018), see Appendix B.13.

**Training and evaluation**    Unless specified, all experiments are repeated for 3 times to report metric means. We use Adam (Kingma & Ba, 2015) and adopt warm up (KL annealing) (Bowman et al., 2016) to train all models. We perform early-stopping using negative log-likelihood (NLL) on validation set, to prevent over-fitting on StaticMNIST and on all datasets with PixelVAE. We use 1,000 samples to estimate NLL and other metrics on test set. See Appendix B.3 for more details.

### 5.2 QUANTITATIVE RESULTS OF VAE WITH REALNVP PRIOR AND ONE LATENT VARIABLE COMPARED TO DEEP HIERARCHICAL VAES AND OTHER PREVIOUS WORKS

In Tables 1 and 2, we compare ResnetVAE and PixelVAE with RealNVP prior to other approaches on StaticMNIST and MNIST. See Tables B.10 and B.11 for results on Omniglot and FashionMNIST, which have a similar trend. All models except ours and that of Huang et al. (2017) used at least 2 latent variables. For comparison between our model and Huang et al. (2017), see Appendix B.7.

Our ResnetVAE with RealNVP prior, $K = 50$ is second only to BIVA among all models without PixelCNN decoder, and ranks the first among all models with PixelCNN decoder. On MNIST, the NLL of our model is very close to BIVA, while the latter used 6 latent variables and very compli-

Table 1: Test NLL on StaticMNIST. "†" indicates a hierarchical model with 2 latent variables, while "‡" indicates at least 3 latent variables.

| Model | NLL |
|---|---|
| *Models without PixelCNN decoder* | |
| ConvHVAE + Lars prior[†] (Bauer & Mnih, 2019) | 81.70 |
| ConvHVAE + VampPrior[†] (Tomczak & Welling, 2018) | 81.09 |
| ResConv + RealNVP prior (Huang et al., 2017) | 81.44 |
| VAE + IAF[‡] (Kingma et al., 2016) | 79.88 |
| BIVA[‡] (Maaløe et al., 2019) | **78.59** |
| **ConvVAE + RNVP** $p(\mathbf{z})$, $K = 50$ | 80.09 |
| **ResnetVAE + RNVP** $p(\mathbf{z})$, $K = 50$ | 79.84 |
| *Models with PixelCNN decoder* | |
| VLAE[‡] (Chen et al., 2017) | 79.03 |
| PixelHVAE + VampPrior[†] (Tomczak & Welling, 2018) | 79.78 |
| **PixelVAE + RNVP** $p(\mathbf{z})$, $K = 50$ | **79.01** |

Table 2: Test NLL on MNIST. "†" and "‡" has the same meaning as Table 1.

| Model | NLL |
|---|---|
| *Models without PixelCNN decoder* | |
| ConvHVAE + Lars prior[†] (Bauer & Mnih, 2019) | 80.30 |
| ConvHVAE + VampPrior[†] (Tomczak & Welling, 2018) | 79.75 |
| VAE + IAF[‡] (Kingma et al., 2016) | 79.10 |
| BIVA[‡] (Maaløe et al., 2019) | **78.41** |
| **ConvVAE + RNVP** $p(\mathbf{z})$, $K = 50$ | 78.61 |
| **ResnetVAE + RNVP** $p(\mathbf{z})$, $K = 50$ | 78.49 |
| *Models with PixelCNN decoder* | |
| VLAE[‡] (Chen et al., 2017) | 78.53 |
| PixelVAE[†] (Gulrajani et al., 2017) | 79.02 |
| PixelHVAE + VampPrior[†] (Tomczak & Welling, 2018) | 78.45 |
| **PixelVAE + RNVP** $p(\mathbf{z})$, $K = 50$ | **78.12** |

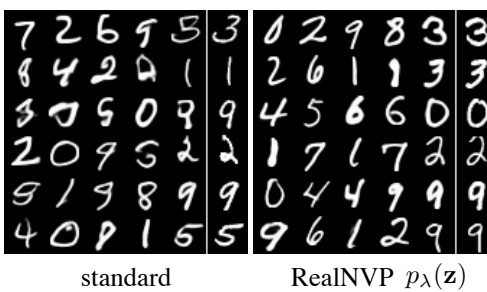

standard        RealNVP $p_\lambda(\mathbf{z})$

Figure 1: Sample means from $p_\lambda(\mathbf{z})$ of ResnetVAE with: (left) Gaussian prior; (right) RealNVP prior. The last column of each 6x6 grid shows the training set images, most similar to the second-to-last column in pixel-wise L2 distance.

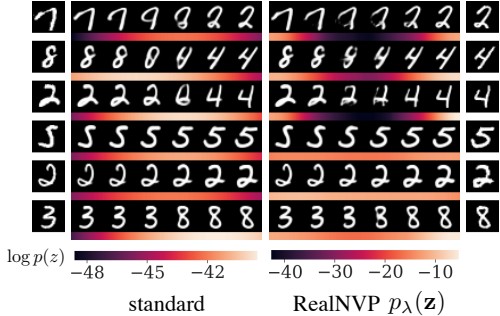

$\log p(z)$  −48  −45  −42    −40  −30  −20  −10
standard        RealNVP $p_\lambda(\mathbf{z})$

Figure 2: Interpolations of $\mathbf{z}$ from ResnetVAE, between the centers of $q_\phi(\mathbf{z}|\mathbf{x})$ of two training points, and heatmaps of $\log p_\lambda(\mathbf{z})$. The left- and right-most columns are the training points.

cated architecture. Meanwhile, our ConvVAE with RealNVP prior, $K = 50$ has lower test NLL than ConvHVAE with *Lars prior* and *VampPrior*. Since ConvVAE is undoubtedly a simpler architecture than ConvHVAE (which has 2 latent variables), it is likely that our improvement comes from the RealNVP prior rather than the different architecture. We also conduct experiments on Cifar10 (Krizhevsky, 2009). Although not as good as state-of-the-art deep VAEs, our ResnetVAE using RealNVP prior with just one latent variable still shows great improvement over the Gaussian prior, in both test bpd and sampling quality. See Appendix B.5 for more details.

**Tables 1 and 2 show that using RealNVP prior with just one latent variable, it is possible to achieve NLLs comparable to very deep state-of-the-art VAE (BIVA), ourperforming many previous works (including works on priors, and works of complicated hierarchical VAE equipped with rich posteriors like VAE + IAF). This discovery shows that shallow VAEs with learned prior and a small number of latent variables is a promising direction.**

## 5.3 QUALITATIVE RESULTS

Figure 1 samples images from ResnetVAE with/without RealNVP prior. Compared to standard ResnetVAE, ResnetVAE with RealNVP prior produces fewer digits that are hard to interpret. The

Table 3: Average test NLL (lower is better) of different models, with Gaussian prior & Gaussian posterior ("standard"), Gaussian prior & RealNVP posterior ("RNVP $q(z|x)$"), and RealNVP prior & Gaussian posterior ("RNVP $p(z)$"). Flow depth $K = 20$.

| Datasets | DenseVAE | | | ResnetVAE | | | PixelVAE | | |
|---|---|---|---|---|---|---|---|---|---|
| | standard | RNVP $q(z\|x)$ | RNVP $p(z)$ | standard | RNVP $q(z\|x)$ | RNVP $p(z)$ | standard | RNVP $q(z\|x)$ | RNVP $p(z)$ |
| StaticMNIST | 88.84 | 86.07 | **84.87** | 82.95 | 80.97 | **79.99** | 79.47 | 79.09 | **78.92** |
| MNIST | 84.48 | 82.53 | **80.43** | 81.07 | 79.53 | **78.58** | 78.64 | 78.41 | **78.15** |
| FashionMNIST | 228.60 | 227.79 | **226.11** | 226.17 | 225.02 | **224.09** | 224.22 | 223.81 | **223.40** |
| Omniglot | 106.42 | 102.97 | **102.19** | 96.99 | 94.30 | **93.61** | 89.83 | 89.69 | **89.61** |

Table 4: Test NLL of ResnetVAE on MNIST, with RealNVP posterior ("$q(z|x)$"), RealNVP prior ("$p(z)$"), and RealNVP prior & posterior ("both"). Flow depth $K$ is $2K_0$ for the posterior or the prior in "$q(z|x)$" and "$p(z)$", while $K_0$ for both the posterior and the prior in "both".

| | $K_0$ | | | |
|---|---|---|---|---|
| ResnetVAE with | 1 | 5 | 10 | 20 |
| $q(z\|x)$, $K = 2K_0$ | 80.29 | 79.68 | 79.53 | 79.49 |
| both, $K = K_0$ | 79.85 | 79.01 | 78.71 | 78.56 |
| $p(z)$, $K = 2K_0$ | **79.58** | **78.75** | **78.58** | **78.51** |

Table 5: Average number of *active units* of ResnetVAE, with standard prior & posterior ("standard"), RealNVP posterior ("RNVP $q(z|x)$"), and RealNVP prior ("RNVP $p(z)$").

| Datasets | ResnetVAE | | |
|---|---|---|---|
| | standard | RNVP $q(z\|x)$ | RNVP $p(z)$ |
| StaticMNIST | 30 | **40** | **40** |
| MNIST | 25.3 | **40** | **40** |
| FashionMNIST | 27 | **64** | **64** |
| Omniglot | 59.3 | **64** | **64** |

last column of each 6x6 grid shows the training set images, most similar to the second-to-last column in pixel-wise L2 distance. There are differences between the last two columns, indicating our model is not just memorizing the training data. More samples are in Appendix B.11.

## 5.4 ABLATION STUDY

**RealNVP prior leads to substantially lower NLLs than standard VAE and RealNVP posterior** Table 3 shows the NLLs of DenseVAE, ResnetVAE and PixelVAE with flow depth $K = 20$. The NLLs of ConvVAE (with similar trends as ResnetVAE) and the standard deviations of all models are reported in Table B.12, while the NLLs of ResnetVAE with different $K$ (range from 1 to 50) can be found in Table B.15. We can see that RealNVP prior consistently outperforms standard VAE and RealNVP posterior in test NLL, with as large improvement as about 2 nats (compared to standard ResnetVAE) or 1 nat (compared to ResnetVAE with RealNVP posterior) on ResnetVAE, and even larger improvement on DenseVAE. The improvement is not so significant on PixelVAE, likely due to the less information encoded in the latent variable of PixelVAE (Gulrajani et al., 2017).

**Using RealNVP prior only has better NLL than using both RealNVP prior and posterior, or using RealNVP posterior only, under the same model complexity**, as shown in Table 4.

**Active units** Table 5 counts the *active units* (Burda et al., 2016) of different ResnetVAEs, which quantifies the number of latent dimensions used for encoding information from input data. We can see that, both RealNVP prior and posterior can make all units of a ResnetVAE to be active (which is in sharp contrast to standard VAE). However, RealNVP prior can in fact result in substantially better *reconstruction loss* than RealNVP posterior (see Table B.18). This indicates that, the good regularization effect, "a learned RealNVP prior can lead to more active units than a fixed prior" (Tomczak & Welling, 2018; Bauer & Mnih, 2019), is not the main cause of the huge improvement in NLLs, especially for the improvement of RealNVP prior over RealNVP posterior.

## 5.5 RECONSTRUCTION LOSS AND POSTERIOR OVERLAPPING

**Better reconstruction loss, but larger KL divergence** In Table 6, *ELBO* and *reconstruction loss* ("*recons*", which is ① in Eq. (5)) of ResnetVAE with RealNVP prior are substantially higher

Table 6: Average test ELBO ("*elbo*"), reconstruction loss ("*recons*"), $\mathbb{E}_{p^\star(\mathbf{x})} D_{\mathrm{KL}}(q_\phi(\mathbf{z}|\mathbf{x})\|p_\lambda(\mathbf{z}))$ ("*kl*"), and $\mathbb{E}_{p^\star(\mathbf{x})} D_{\mathrm{KL}}(q_\phi(\mathbf{z}|\mathbf{x})\|p_\theta(\mathbf{z}|\mathbf{x}))$ ("*$kl_{z|x}$*") of ResnetVAE with different priors.

| | **standard** | | | | **RealNVP** $p(z)$ | | | |
|---|---|---|---|---|---|---|---|---|
| **Datasets** | *elbo* | *recons* | *kl* | *$kl_{z\|x}$* | *elbo* | *recons* | *kl* | *$kl_{z\|x}$* |
| StaticMNIST | -87.61 | -60.09 | **27.52** | 4.67 | **-82.85** | **-54.32** | 28.54 | **2.87** |
| MNIST | -84.62 | -58.70 | **25.92** | 3.55 | **-80.34** | **-53.64** | 26.70 | **1.76** |
| FashionMNIST | -228.91 | -208.94 | **19.96** | 2.74 | **-225.97** | **-204.66** | 21.31 | **1.88** |
| Omniglot | -104.87 | -66.98 | **37.89** | 7.88 | **-99.60** | **-61.21** | 38.39 | **5.99** |

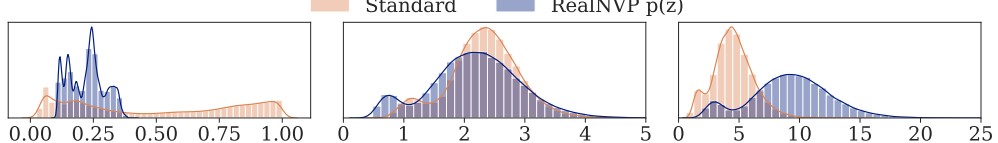

Figure 3: Histograms of: (left) per-dimensional stds of $q_\phi(\mathbf{z}|\mathbf{x})$; (middle) distances between closest pairs of $q_\phi(\mathbf{z}|\mathbf{x})$; and (right) *normalized distances*. See Appendix B.4 for formulation.

than standard ResnetVAE, just as the trend of test log-likelihood (LL) in Table 3. Metrics of other models are in Tables B.16 to B.19. On the contrary, $\mathbb{E}_{p^\star(\mathbf{x})} D_{\mathrm{KL}}(q_\phi(\mathbf{z}|\mathbf{x})\|p_\lambda(\mathbf{z}))$ ("*kl*", which is ② + ③) are larger. Since ELBO equals to ① − (② + ③), this suggests that in our experiments, the improvement in ELBO (and also NLL) of ResnetVAE with RealNVP prior all comes from the improved reconstruction loss. This serves as a foundation for the analysis in Section 4.

*kl* can happen to be smaller when *recons* gets larger (*e.g.*, our DenseVAE in Table B.16, and VAE with dense layers of (Bauer & Mnih, 2019, Table 2)). Nevertheless, all our experiments with Real-NVP prior show improved reconstruction loss, which validates the analysis of Section 4.

**Smaller standard deviation of Gaussian posterior with RealNVP prior**   In Fig. 3, we plot the histograms of per-dimensional stds of $q_\phi(\mathbf{z}|\mathbf{x})$, as well as the distances and *normalized distances* (which is roughly distance/std) between each closest pair of $q_\phi(\mathbf{z}|\mathbf{x})$ (see Appendix B.4 for formulations). The stds of $q_\phi(\mathbf{z}|\mathbf{x})$ with RealNVP prior are substantially smaller, and the *normalized distances* are larger. Larger *normalized distances* indicate less density of $q_\phi(\mathbf{z}|\mathbf{x})$ to be overlapping. This fact is a direct evidence of the analysis in Section 4.

**The test NLL of the model and the learned std of $q_\phi(\mathbf{z}|\mathbf{x})$ is insensitive to the minimum clipping value under a certain threshold**   See Appendix B.6 for experiment results.

**Appropriate overlapping among $q_\phi(\mathbf{z}|\mathbf{x})$ with learned prior**   To demonstrate that the stds of $q_\phi(\mathbf{z}|\mathbf{x})$ with RealNVP prior are reduced according to the dissimilarity between $\mathbf{x}$ rather than being reduced equally (*i.e.*, $q_\phi(\mathbf{z}|\mathbf{x})$ exhibits "appropriate overlapping"), we plot the interpolations of $\mathbf{z}$ between the centers of $q_\phi(\mathbf{z}|\mathbf{x})$ of two training points, and $\log p_\lambda(\mathbf{z})$ of these interpolations in Fig. 2, We visualize $p_\lambda(\mathbf{z})$, because it is trained to match $q_\phi(\mathbf{z})$, and can be computed much more reliable than $q_\phi(\mathbf{z})$; and because the density of $q_\phi(\mathbf{z})$ between $\mathbf{z}$ corresponding to two $\mathbf{x}$ points can be an indicator of how $q_\phi(\mathbf{z}|\mathbf{x})$ overlap between them. The learned RealNVP $p_\lambda(\mathbf{z})$ scores the interpolations of $\mathbf{z}$ between the centers of $q_\phi(\mathbf{z}|\mathbf{x})$ of two training points, giving low likelihoods to hard-to-interpret interpolated samples between two dissimilar $\mathbf{x}$ (the first three rows), while giving high likelihoods to good quality samples between two similar $\mathbf{x}$ (the last three rows). In contrast, the unit Gaussian prior assigns high likelihoods to all interpolations, even to hard-to-interpret ones. This suggests that the posterior overlapping is more appropriate with RealNVP prior than with unit Gaussian prior.

**Learned prior influences the trade-off between reconstruction loss and KL divergence**, as suggested by Section 4. We plot the rate-distortion curve (RD curve) (Alemi et al., 2018) of $\beta$-ResnetVAE trained with different $\beta$ and prior flow depth $K$ in Fig. 4. Rate is $D_{\mathrm{KL}}(q_\phi(\mathbf{z}|\mathbf{x})\|p_\theta(\mathbf{z}))$, while distortion is negative reconstruction loss. Each connected curve with the same shape of points in Fig. 4 correspond to the models with the same $K$, but different $\beta$. We can see that the curves of $K = 1$ is closer to the boundary formed by the green line and the x & y axes than $K = 0$, while $K = 20$ & $50$ are even closer. According to Alemi et al. (2018), points on the RD curve being closer

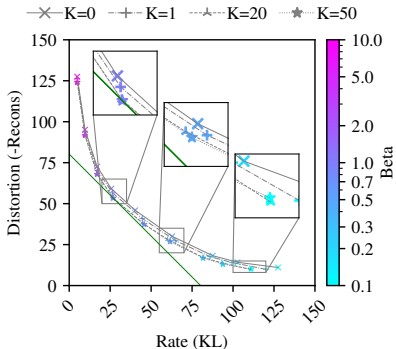
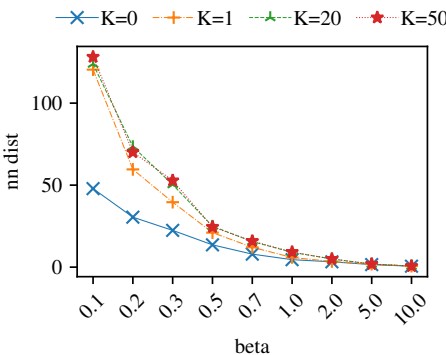

Figure 4: Rate ($D_{\mathrm{KL}}(q_\phi(\mathbf{z}|\mathbf{x})\|p_\theta(\mathbf{z}))$) and distortion (-*reconstruction loss*) of $\beta$-ResnetVAE trained with different $\beta$ and prior flow depth $K$.

Figure 5: Average *normalized distance* of $\beta$-ResnetVAE trained with different $\beta$ and prior flow depth $K$.

to the boundary suggests that the corresponding models are closer to the theoretical optimal models on a particular dataset, when traded between reconstruction loss and KL divergence. Given this, we conclude that learned prior can lead to a "better" trade-off from the perspective of RD curve.

We also plotted the average *normalized distance* of $\beta$-ResnetVAE trained with different $\beta$ and prior flow depth $K$ in Fig. 5. The average *std* of $q_\phi(\mathbf{z}|\mathbf{x})$ can be found at Fig. B.6. Learned prior can encourage less posterior overlapping than unit Gaussian prior for various $\beta$, not only for $\beta = 1$.

# 6 RELATED WORK

Learned priors, as a natural choice for the conditional priors of intermediate variables, have long been unintentionally used in hierarchical VAEs (Rezende et al., 2014; Sønderby et al., 2016; Kingma et al., 2016; Maaløe et al., 2019). A few works were proposed to enrich the prior, *e.g.*, Gaussian mixture priors (Nalisnick et al., 2016; Dilokthanakul et al., 2016), Bayesian non-parametric priors (Nalisnick & Smyth, 2017; Goyal et al., 2017), and auto-regressive priors (Gulrajani et al., 2017; Chen et al., 2017), without the awareness of its relationship with the *aggregated posterior*, until the analysis made by Hoffman & Johnson (2016). Since then, attempts have been made in matching the prior to *aggregated posterior*, by using RealNVP (Huang et al., 2017), variational mixture of posteriors (Tomczak & Welling, 2018), learned accept/reject sampling (Bauer & Mnih, 2019), and kernel density trick (Takahashi et al., 2019). However, none of these works recognized the improved reconstruction loss induced by learned prior. Furthermore, they did not show that learned prior with just one latent variable can achieve comparable results to those of many deep hierarchical VAEs.

The trade-off between reconstruction loss and KL divergence was discussed in the context of $\beta$-VAE (Higgins et al., 2017; Alemi et al., 2018; Mathieu et al., 2019), however, they did not further discuss the impact of a learned prior on this trade-off. Mathieu et al. (2019) also discussed the posterior overlapping, but only within the $\beta$-VAE framework, thus was only able to control the degree of overlapping globally, without considering the local dissimilarity between $\mathbf{x}$.

# 7 CONCLUSION

In this paper, using learned RealNVP prior with just one latent variable in VAE, we managed to achieve test NLLs comparable to very deep state-of-the-art hierarchical VAE, outperforming many previous works of complex hierarchical VAEs equipped with rich priors/posteriors. We hypothesized that, when coupled with Gaussian posteriors, the learned prior can encourage appropriate posterior overlapping, which is likely to benefit the reconstruction loss and lower-bound, supported by our experimental results. We showed that with learned RealNVP prior, $\beta$-VAE can have better rate-distortion curve (Alemi et al., 2018) than with fixed Gaussian prior. We believe this paper is an important step towards shallow VAEs with learned prior and a small number of latent variables, which potentially can be more scalable to large datasets than those deep hierarchical VAEs.

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

# A    PROOF FOR PROPOSITION 1

We first re-state Proposition 1:

**Proposition.** *Given a finite number of discrete training data, i.e., $p^\star(\mathbf{x}) = \frac{1}{N}\sum_{i=1}^{N}\delta(\mathbf{x} - \mathbf{x}^{(i)})$, if $p_\theta(\mathbf{x}|\mathbf{z}) = \mathrm{Bernoulli}(\boldsymbol{\mu}_\theta(\mathbf{z}))$, where the Bernoulli mean $\boldsymbol{\mu}_\theta(\mathbf{z})$ is produced by the decoder, and $0 < \mu_\theta^k(\mathbf{z}) < 1$ for each of its $k$-th dimensional output, then the optimal decoder $\boldsymbol{\mu}_\theta(\mathbf{z})$ is:*

$$\boldsymbol{\mu}_\theta(\mathbf{z}) = \sum_i w_i(\mathbf{z})\,\mathbf{x}^{(i)}, \quad \text{where } w_i(\mathbf{z}) = \frac{q_\phi(\mathbf{z}|\mathbf{x}^{(i)})}{\sum_j q_\phi(\mathbf{z}|\mathbf{x}^{(j)})} \text{ and } \sum_i w_i(\mathbf{z}) = 1$$

We shall need the Euler's equation for the calculus of variations (see Gelfand & Silverman (2000), page 15, 24, and 35)):

**Theorem A.1.** *Let $\mathcal{L}[f]$ be a functional of the form:*

$$\mathcal{L}[f] = \int_a^b F(x, f, \partial_x f)\, dx$$

*defined on the set of functions $f(x)$ which satisfies the boundary conditions $y(a) = A$, $y(b) = B$, then a necessary condition for $\mathcal{L}[f]$ to have an extremum for a given function $f(x)$ is that, $f(x)$ satisfy Euler's equation (Gelfand & Silverman, 2000, page 15):*

$$\frac{\partial F}{\partial f} - \frac{d}{dx}\frac{\partial F}{\partial(\partial_x f)} = 0 \tag{A.1}$$

*where we use $\partial_x f$ to denote $\frac{df}{dx}$.*

*For multi-variate function $f(x_1, x_2)$ and the functional:*

$$\mathcal{L}[f] = \iint_R F(x_1, x_2, f, \partial_{x_1} f, \partial_{x_2} f)\, dx_1\, dx_2$$

*where $R$ is a closed area, the Euler's equation becomes (Gelfand & Silverman, 2000, page 24):*

$$\frac{\partial F}{\partial f} - \frac{\partial}{\partial x_1}\frac{\partial F}{\partial(\partial_{x_1} f)} - \frac{\partial}{\partial x_2}\frac{\partial F}{\partial(\partial_{x_2} f)} = 0 \tag{A.2}$$

*For vector value function $f(x) = [f^1(x), \ldots, f^n(x)]$ and the functional:*

$$\mathcal{L}[f] = \int_a^b F(x, f^1, \ldots, f^n, \partial_x f^1, \ldots, \partial_x f^n)\, dx$$

*a necessary condition for $f(x)$ to be an extremal of $\mathcal{L}[f]$ is that, every $f^i(x)$ satisfies Euler's equation (Gelfand & Silverman, 2000, page 35).*

With some straightforward extensions, we can also get to the Euler's equation for multi-variate vector value functions.

The original Euler's equation requires the integral to be definite, whereas a neural network is defined on the whole $\mathbb{R}^n$ space. However, in practice, the input values (*i.e.*, the training and test data) are usually within limited range, and the norm of the neural network parameters are not too large. As a result, there should exist a sufficiently large number $R$, such that all intermediate outputs are within the closed area $[-R, R]^n$. In this sense, the definite and indefinite integral would not have too much difference. We shall always use the indefinite integral notations in the following proof.

Finally, we get to the proof of Proposition 1:

*Proof.* To apply calculus of variations, we need to substitute the parameterized, bounded $\boldsymbol{\mu}_\theta(\mathbf{z})$ with a non-parameterized, unbounded mapping. Since $0 < \mu_\theta^k(\mathbf{z}) < 1$, and $\boldsymbol{\mu}_\theta(\mathbf{z})$ is produced by neural network, which ensures $\boldsymbol{\mu}_\theta(\mathbf{z})$ is a continuous mapping, then $\forall\theta$, there exists unbounded $\mathbf{t}(\mathbf{z})$, *s.t.*

$$\mu_\theta^k(\mathbf{z}) = \frac{\exp(t^k(\mathbf{z}))}{1 + \exp(t^k(\mathbf{z}))}$$

$$t^k(\mathbf{z}) = \log\mu_\theta^k(\mathbf{z}) - \log(1 - \mu_\theta^k(\mathbf{z}))$$

and for all continuous mapping $\mathbf{t}(\mathbf{z})$, there also exists $\boldsymbol{\mu}_\theta(\mathbf{z})$, satisfying the above equations. In fact, this substitution is also adopted in the actual implementation of our models.

The probability of $p_\theta(\mathbf{x}|\mathbf{z})$ is given by:

$$p_\theta(\mathbf{x}|\mathbf{z}) = \mathrm{Bernoulli}(\boldsymbol{\mu}_\theta(\mathbf{z})) = \prod_k \left(\mu_\theta^k(\mathbf{z})\right)^{x_k} \left(1 - \mu_\theta^k(\mathbf{z})\right)^{(1-x_k)}$$

Then we have:

$$
\begin{aligned}
\log p_\theta(\mathbf{x}|\mathbf{z}) &= \sum_k \left\{ x_k \log \mu_\theta^k(\mathbf{z}) + (1-x_k)\log\left(1-\mu_\theta^k(\mathbf{z})\right) \right\} \\
&= \sum_k \left\{ x_k \, t^k(\mathbf{z}) - x_k \log\left[1+\exp\!\left(t^k(\mathbf{z})\right)\right] - (1-x_k)\log\left[1+\exp\!\left(t^k(\mathbf{z})\right)\right] \right\} \\
&= \sum_k \left\{ x_k \, t^k(\mathbf{z}) - \log\left[1+\exp\!\left(t^k(\mathbf{z})\right)\right] \right\}
\end{aligned}
$$

The training objective $\mathcal{L}$ can be then formulated as a functional on $\mathbf{t}(\mathbf{z})$:

$$
\begin{aligned}
\mathcal{L}[\mathbf{t}] &= \mathbb{E}_{p^\star(\mathbf{x})} \mathbb{E}_{q_\phi(\mathbf{z}|\mathbf{x})} \left[ \log p_\theta(\mathbf{x}|\mathbf{z}) + \log\frac{p_\theta(\mathbf{z})}{q_\phi(\mathbf{z}|\mathbf{x})} \right] \\
&= \iint p^\star(\mathbf{x})\, q_\phi(\mathbf{z}|\mathbf{x}) \cdot \left( \log p_\theta(\mathbf{x}|\mathbf{z}) + \log\frac{p_\theta(\mathbf{z})}{q_\phi(\mathbf{z}|\mathbf{x})} \right) \mathrm{d}\mathbf{z}\,\mathrm{d}\mathbf{x} \\
&= \int F(\mathbf{z},\mathbf{t})\,\mathrm{d}\mathbf{z}
\end{aligned}
$$

where $F(\mathbf{z},\mathbf{t})$ is:

$$F(\mathbf{z},\mathbf{t}) = \int p^\star(\mathbf{x})\, q_\phi(\mathbf{z}|\mathbf{x}) \left[ \sum_k \left\{ x_k \, t^k(\mathbf{z}) - \log\left[1+\exp\!\left(t^k(\mathbf{z})\right)\right] \right\} + \log\frac{p_\theta(\mathbf{z})}{q_\phi(\mathbf{z}|\mathbf{x})} \right] \mathrm{d}\mathbf{x}$$

According to Euler's equation (Gelfand & Silverman, 2000, page 14 and 35), the necessary condition for $\mathcal{L}[\mathbf{t}]$ to have an extremum for a given $\mathbf{t}(\mathbf{z})$ is that, $\mathbf{t}(\mathbf{z})$ satisfies $\partial F/\partial t^k = 0, \forall k$. Thus we have:

$$
\begin{aligned}
\frac{\partial F}{\partial t^k} = 0 &\implies \int p^\star(\mathbf{x})\, q_\phi(\mathbf{z}|\mathbf{x}) \left[ x_k - \frac{\exp\!\left(t^k(\mathbf{z})\right)}{1+\exp\!\left(t^k(\mathbf{z})\right)} \right] \mathrm{d}\mathbf{x} = 0 \\
&\implies \int p^\star(\mathbf{x})\, q_\phi(\mathbf{z}|\mathbf{x}) \left[ x_k - \mu_\theta^k(\mathbf{z}) \right] \mathrm{d}\mathbf{x} = 0 \\
&\implies \sum_i q_\phi(\mathbf{z}|\mathbf{x}^{(i)}) \left[ x_k^{(i)} - \mu_\theta^k(\mathbf{z}) \right] = 0 \\
&\implies \sum_i q_\phi(\mathbf{z}|\mathbf{x}^{(i)})\, x_k^{(i)} = \left( \sum_i q_\phi(\mathbf{z}|\mathbf{x}^{(i)}) \right) \mu_\theta^k(\mathbf{z}) \\
&\implies \mu_\theta^k(\mathbf{z}) = \frac{\sum_i q_\phi(\mathbf{z}|\mathbf{x}^{(i)})\, x_k^{(i)}}{\sum_j q_\phi(\mathbf{z}|\mathbf{x}^{(j)})}
\end{aligned}
$$

That is to say, $\boldsymbol{\mu}_\theta(\mathbf{z}) = \frac{\sum_i q_\phi(\mathbf{z}|\mathbf{x}^{(i)})\, \mathbf{x}^{(i)}}{\sum_j q_\phi(\mathbf{z}|\mathbf{x}^{(j)})} = \sum_i w_i(\mathbf{z})\, \mathbf{x}^{(i)}$. $\qquad\square$

Rezende & Viola (2018) has proved that when $p_\theta(\mathbf{x}|\mathbf{z}) = \mathcal{N}(\boldsymbol{\mu}_\theta(\mathbf{z}), \sigma^2\mathbf{I})$, where $\sigma$ is a global fixed constant, the optimal decoder $\boldsymbol{\mu}_\theta(\mathbf{z}) = \sum_i w_i(\mathbf{z})\, \mathbf{x}^{(i)}$, which is exactly the same as our conclusion. Rosca et al. (2018) has proved that the gradient of $\mathrm{Bernoulli}(\boldsymbol{\mu}_\theta(\mathbf{z}))$ is the same as that of $\mathcal{N}(\boldsymbol{\mu}_\theta(\mathbf{z}), \sigma^2\mathbf{I})$ when $\sigma = 1$, but they did not calculate out the optimal decoder. We push forward both these works.

# B EXPERIMENTAL DETAILS

## B.1 DATASETS

**MNIST** MNIST is a 28x28 grayscale image dataset of hand-written digits, with 60,000 data points for training and 10,000 for testing. When validation is required for early-stopping, we randomly split the training data into 50,000 for training and 10,000 for validation.

Since we use Bernoulli $p_\theta(\mathbf{x}|\mathbf{z})$ to model these images in VAE, we binarize these images by the method in (Salakhutdinov & Murray, 2008): each pixel value is randomly set to 1 in proportion to its pixel intensity. The training and validation images are re-binarized at each epoch. However, the test images are binarized beforehand for all experiments. We binarize each test image 10 times, and use all these 10 binarized data points in evaluation. This method results in a 10 times larger test set, but we believe this can help us to obtain a more objective evaluation result.

**StaticMNIST** StaticMNIST (Larochelle & Murray, 2011) is a pre-binarized MNIST image dataset, with the original 60,000 training data already splitted into 50,000 for training and 10,000 for validation. We always use validation set for early-stopping on StaticMNIST. Meanwhile, since StaticMNIST has already been binarized, the test set is used as-is without 10x enlargement.

**FashionMNIST** FashionMNIST (Xiao et al., 2017) is a recently proposed image dataset of grayscale fashion products, with the same specification as MNIST. We thus use the same training-validation split and the same binarization method just as MNIST.

**Omniglot** Omniglot (Lake et al., 2015) is a 28x28 grayscale image dataset of hand-written characters. We use the preprocessed data from (Burda et al., 2016), with 24,345 data points for training and 8,070 for testing. When validation is required, we randomly split the training data into 20,345 for training and 4,000 for validation. We use dynamic binarization on Omniglot just as MNIST.

## B.2 NETWORK ARCHITECTURES

**Notations** In order to describe the detailed architecture of our models, we will introduce auxiliary functions to denote network components. A function $h_\phi(\mathbf{x})$ should denote a sub-network in $q_\phi(\mathbf{z}|\mathbf{x})$, with a subset of $\phi$ as its own learnable parameters. For example, if we write $q_\phi(\mathbf{z}|\mathbf{x}) = q_\phi(\mathbf{z}|h_\phi(\mathbf{x})) = \mathcal{N}(\boldsymbol{\mu}_\phi(h_\phi(\mathbf{x})), \boldsymbol{\sigma}_\phi^2(h_\phi(\mathbf{x}))\,\mathbf{I})$, it means that the the posterior $q_\phi(\mathbf{z}|\mathbf{x})$ is a Gaussian, whose mean and standard deviation are derived by one shared sub-network $h_\phi(\mathbf{x})$ and two separate sub-networks $\boldsymbol{\mu}_\phi(\cdot)$ and $\boldsymbol{\sigma}_\phi(\cdot)$, respectively.

The structure of a network is described by composition of elementary neural network layers.

Linear[$k$] indicates a linear dense layer with $k$ outputs. Dense[$k$] indicates a non-linear dense layer. $a \to b$ indicates a composition of $a$ and $b$, *e.g.*, $Dense[m] \to Dense[n]$ indicates two successive dense layers, where the first layer has $m$ outputs and the second layer has $n$ outputs. These two dense layers can also be abbreviated as Dense[$m \to n$].

Conv[$H \times W \times C$] denotes a non-linear convolution layer, whose output shape is $H \times W \times C$, where $H$ is the height, $W$ is the width and $C$ is the channel size. As an abbreviation, Conv[$H_1 \times W_1 \times C_1 \to H_2 \times W_2 \times C_2$] denotes two successive non-linear convolution layers. Resnet[$\cdot$] denotes non-linear resnet layer(s) (Zagoruyko & Komodakis, 2016). All Conv and Resnet layers by default use 3x3 kernels, unless the kernel size is specified as subscript (*e.g.*, $\text{Conv}_{1\times1}$ denotes a 1x1 convolution layer). The strides of Conv and Resnet layers are automatically determined by the input and output shapes, which is 2 in most cases. LinearConv[$\cdot$] denotes linear convolution layer(s).

DeConv[$\cdot$] and DeResnet[$\cdot$] denote deconvolution and deconvolutional resnet layers, respectively.

PixelCNN[$\cdot$] is a PixelCNN layer proposed by Salimans et al. (2017). It uses resnet layers, instead of convolution layers. Details can be found in its original paper.

Flatten indicates to reshape the input 3-d tensor into a vector, while UnFlatten[$H \times W \times C$] indicates to reshape the input vector into a 3-d tensor of shape $H \times W \times C$. Concat[$a, b$] indicates to concat the output of $a$ and $b$ along the last axis.

CouplingLayer and ActNorm are components of RealNVP, proposed by Dinh et al. (2017); Kingma & Dhariwal (2018). InvertibleDense is a component modified from *invertible 1x1 convolution* (Kingma & Dhariwal, 2018). We shall only introduce the details of CouplingLayer, since it contains sub-networks, while the rest two are just simple components.

**General configurations**  All non-linear layers use *leaky relu* (Maas et al., 2013) activation.

The observed variable $\mathbf{x}$ (*i.e.*, the input image) is always a 3-d tensor, with shape $H \times W \times C$, whether or not the model is convolutional. The latent variable $\mathbf{z}$ is a vector, whose number of dimensions is chosen to be 40 on MNIST and StaticMNIST, while 64 on FashionMNIST and Omniglot. This is because we think the latter two datasets are conceptually more complicated than the former two, thus requiring higher dimensional latent variables.

The Gaussian posterior $q_\phi(\mathbf{z}|\mathbf{x})$ is derived as:

$$q_\phi(\mathbf{z}|\mathbf{x}) = \mathcal{N}(\boldsymbol{\mu}_\phi(h_\phi(\mathbf{x})), \boldsymbol{\sigma}_\phi^2(h_\phi(\mathbf{x}))\,\mathbf{I})$$
$$\boldsymbol{\mu}_\phi(h_\phi(\mathbf{x})) = h_\phi(\mathbf{x}) \to \text{Linear}[\text{Dim}(\mathbf{z})]$$
$$\log\boldsymbol{\sigma}_\phi(h_\phi(\mathbf{x})) = h_\phi(\mathbf{x}) \to \text{Linear}[\text{Dim}(\mathbf{z})]$$

Note we make the network to produce $\log\boldsymbol{\sigma}_\phi(h_\phi(\mathbf{x}))$ instead of directly producing $\boldsymbol{\sigma}_\phi(h_\phi(\mathbf{x}))$. $h_\phi(\mathbf{x})$ is the hidden layers, varying among different models.

For binarized images, we use Bernoulli conditional distribution $p_\theta(\mathbf{x}|\mathbf{z})$, derived as:

$$p_\theta(\mathbf{x}|\mathbf{z}) = \text{Bernoulli}[\boldsymbol{\mu}_\theta(h_\theta(\mathbf{z}))]$$

For DenseVAE:

$$\log\frac{\boldsymbol{\mu}_\theta(h_\theta(\mathbf{z}))}{1 - \boldsymbol{\mu}_\theta(h_\theta(\mathbf{z}))} = h_\theta(\mathbf{z}) \to \text{Linear}[784] \to \text{UnFlatten}[28 \times 28 \times 1]$$

And for others (*i.e.*, ConvVAE, ResnetVAE and PixelVAE):

$$\log\frac{\boldsymbol{\mu}_\theta(h_\theta(\mathbf{z}))}{1 - \boldsymbol{\mu}_\theta(h_\theta(\mathbf{z}))} = h_\theta(\mathbf{z}) \to \text{LinearConv}_{1\times1}[28 \times 28 \times 1]$$

Note we make the network to produce $\log\frac{\boldsymbol{\mu}_\theta(h_\theta(\mathbf{z}))}{1 - \boldsymbol{\mu}_\theta(h_\theta(\mathbf{z}))}$, the *logits* of Bernoulli distribution, instead of producing the Bernoulli mean $\boldsymbol{\mu}_\theta(h_\theta(\mathbf{z}))$ directly.

**DenseVAE**  $h_\phi(\mathbf{x})$ and $h_\theta(\mathbf{z})$ of DenseVAE are composed of dense layers, formulated as:

$$h_\phi(\mathbf{x}) = \mathbf{x} \to \text{Flatten} \to \text{Dense}[500 \to 500]$$
$$h_\theta(\mathbf{z}) = \mathbf{z} \to \text{Dense}[500 \to 500]$$

**Conv/ResnetVAE**  $h_\phi(\mathbf{x})$ and $h_\theta(\mathbf{z})$ of ConvVAE are composed of (de)convolutional layers, while those of ResnetVAE consist of (deconvolutional) resnet layers. We only describe the architecture of ResnetVAE here. The structure of ConvVAE can be easily obtained by replacing all (deconvolutional) resnet layers with (de)convolution layers:

$$\begin{aligned}
h_\phi(\mathbf{x}) = \mathbf{x} \to \text{Resnet}[&28 \times 28 \times 32 \to 28 \times 28 \times 32 \\
&\to 14 \times 14 \times 64 \to 14 \times 14 \times 64 \\
&\to 7 \times 7 \times 64 \to 7 \times 7 \times 16] \\
\to \text{Flatten}& \\
h_\theta(\mathbf{z}) = \mathbf{z} \to \text{Dense}[784] &\to \text{UnFlatten}[7 \times 7 \times 16] \\
\to \text{DeResnet}[&7 \times 7 \times 64 \to 14 \times 14 \times 64 \\
&\to 14 \times 14 \times 64 \to 28 \times 28 \times 32 \\
&\to 28 \times 28 \times 32]
\end{aligned}$$

**PixelVAE**   $h_\phi(\mathbf{x})$ of PixelVAE is exactly the same as ResnetVAE, while $h_\theta(\mathbf{z})$ is derived as:

$$
\begin{aligned}
h_\theta(\mathbf{z}) &= \text{Concat}[\mathbf{x}, \tilde{h}_\theta(\mathbf{z})] \\
&\rightarrow \text{PixelCNN}[28 \times 28 \times 33 \rightarrow 28 \times 28 \times 33 \\
&\qquad\rightarrow 28 \times 28 \times 33] \\
\tilde{h}_\theta(\mathbf{z}) &= \mathbf{z} \rightarrow \text{Dense}[784] \rightarrow \text{UnFlatten}[7 \times 7 \times 16] \\
&\rightarrow \text{DeResnet}[7 \times 7 \times 64 \rightarrow 14 \times 14 \times 64 \\
&\qquad\rightarrow 14 \times 14 \times 64 \rightarrow 28 \times 28 \times 32 \\
&\qquad\rightarrow 28 \times 28 \times 32]
\end{aligned}
$$

As Salimans et al. (2017), we use dropout in PixelCNN layers, with rate 0.5.

**RealNVP**   The RealNVP consists of $K$ blocks, while each block consists of an *invertible dense*, a *coupling layer*, and an *actnorm*. The RealNVP mapping for prior, *i.e.* $f_\lambda(\mathbf{z})$, can be formulated as:

$$
\begin{aligned}
f_\lambda(\mathbf{z}) &= \mathbf{z} \rightarrow f_1(\mathbf{h}_1) \rightarrow \cdots \rightarrow f_K(\mathbf{h}_K) \\
f_k(\mathbf{h}_k) &= \mathbf{h}_k \rightarrow \text{InvertibleDense} \rightarrow \text{CouplingLayer} \rightarrow \text{ActNorm} \\
\text{CouplingLayer}(\mathbf{u}) &= \text{Concat}\left[\mathbf{u}_l, \mathbf{u}_r \odot \text{Sigmoid}\left(s_\lambda(h_{\lambda,k}(\mathbf{u}_l))\right) + t_\lambda(h_{\lambda,k}(\mathbf{u}_l))\right] \\
h_{\lambda,k}(\mathbf{u}_l) &= \mathbf{u}_l \rightarrow \text{Dense}[256] \\
s_\lambda(h_{\lambda,k}(\mathbf{u}_l)) &= h_{\lambda,k}(\mathbf{u}_l) \rightarrow \text{Linear}[\text{Dim}(\mathbf{u}_r)] \\
t_\lambda(h_{\lambda,k}(\mathbf{u}_l)) &= h_{\lambda,k}(\mathbf{u}_l) \rightarrow \text{Linear}[\text{Dim}(\mathbf{u}_r)]
\end{aligned}
$$

where $\mathbf{u}_l = \mathbf{u}_{0:\lfloor \text{Dim}(\mathbf{u})/2 \rfloor}$ is the left half of $\mathbf{u}$, and $\mathbf{u}_r = \mathbf{u}_{\lfloor \text{Dim}(\mathbf{u})/2 \rfloor:\text{Dim}(\mathbf{u})}$ is the right half.

The RealNVP posterior, derived from the original Gaussian posterior $q_\phi(\mathbf{w}|\mathbf{x})$, is denoted as $q_{\phi,\eta}(\mathbf{z}|\mathbf{x})$, and formulated as:

$$
q_{\phi,\eta}(\mathbf{z}|\mathbf{x}) = q_\phi(\mathbf{w}|\mathbf{x}) \left| \frac{\partial f_\eta(\mathbf{w})}{\partial \mathbf{w}} \right|^{-1} \tag{B.1}
$$
$$
\mathbf{z} = f_\eta(\mathbf{w})
$$

where the structure of the RealNVP mapping $f_\eta(\mathbf{w})$ for posterior is exactly the same as $f_\lambda(\mathbf{z})$ for prior. The ELBO for VAE with RealNVP posterior is then simply:

$$
\begin{aligned}
\mathcal{L}(\lambda, \theta, \phi, \eta) = \mathbb{E}_{p^\star(\mathbf{x})} \mathbb{E}_{q_\phi(\mathbf{w}|\mathbf{x})} \bigg[ &\log p_\theta(\mathbf{x}|f_\eta(\mathbf{w})) + \log p_\lambda(f_\eta(\mathbf{w})) \\
&- \log q_\phi(\mathbf{w}|\mathbf{x}) + \log \left| \det\left(\frac{\partial f_\eta(\mathbf{w})}{\partial \mathbf{w}}\right) \right| \bigg]
\end{aligned} \tag{B.2}
$$

### B.3   Additional details of training and evaluation

**General methodology**   All the mathematical expressions of expectations *w.r.t.* some distributions are computed by Monte Carlo integration. For example, $\mathbb{E}_{q_\phi(\mathbf{z}|\mathbf{x})}[f(\mathbf{z}, \mathbf{x})]$ is estimated by:

$$
\mathbb{E}_{q_\phi(\mathbf{z}|\mathbf{x})}[f(\mathbf{z}, \mathbf{x})] \approx \frac{1}{L} \sum_{i=1}^{L} f(\mathbf{z}^{(i)}, \mathbf{x})
$$

where $\mathbf{z}^{(i)}$ is one sample from $q_\phi(\mathbf{z}|\mathbf{x})$.

**Training**   We use Adam (Kingma & Ba, 2015) to train our models for 2,400 epochs. The batch size is 128 for DenseVAE, ConvVAE and ResnetVAE, and 64 for PixelVAE. On MNIST, FashionMNIST and Omniglot, we set the learning rate to be $10^{-3}$ in the first 800 epochs, $10^{-4}$ in the next 800 epochs, and $10^{-5}$ in the last 800 epochs. On StaticMNIST, we set the learning rate to be $10^{-4}$ in the first 1,600 epochs, and $10^{-5}$ in the last 800 epochs.

L2 regularization with factor $10^{-4}$ is applied on weights of all non-linear hidden layers, *i.e.*, kernels of non-linear dense layers and convolutional layers, in $h_\phi(\mathbf{x})$, $h_\theta(\mathbf{z})$, $f_\lambda(\mathbf{z})$ and $f_\eta(\mathbf{w})$.

The ELBO is estimated by 1 $\mathbf{z}$ sample for each $\mathbf{x}$ in training. We adopt warm-up (KL annealing) (Bowman et al., 2016). The ELBO using warm-up is formulated as:

$$\mathcal{L}(\mathbf{x}; \lambda, \theta, \phi) = \mathbb{E}_{q_\phi(\mathbf{z}|\mathbf{x})} \left[ \log p_\theta(\mathbf{x}|\mathbf{z}) + \beta \left( \log p_\lambda(\mathbf{z}) - \log q_\phi(\mathbf{z}|\mathbf{x}) \right) \right]$$

$\beta$ is increased from 0.01 to 1 linearly in the first 100 epochs, and it remains 1 afterwards. The warm-up ELBO for VAEs with RealNVP priors and posteriors can be obtained by replacing $p_\lambda(\mathbf{z})$ and $q_\phi(\mathbf{z}|\mathbf{x})$ of the above equation by Eq. (6) and Eq. (B.1), respectively.

We adopt early-stopping using NLL on validation set, to prevent over-fitting on StaticMNIST and PixelVAE. The validation NLL is estimated using 100 $\mathbf{z}$ samples for each $\mathbf{x}$, every 20 epochs.

**Training strategies for $p_\lambda(\mathbf{z})$**   We consider three training strategies for optimizing Eq. (7):

- **Post-hoc training** (Bauer & Mnih, 2019): $q_\phi(\mathbf{z}|\mathbf{x})$ and $p_\theta(\mathbf{x}|\mathbf{z})$ are firstly trained *w.r.t.* the unit Gaussian prior, then $q_\phi(\mathbf{z}|\mathbf{x})$ and $p_\theta(\mathbf{x}|\mathbf{z})$ are fixed and $p_\lambda(\mathbf{z})$ is in turn optimized. This is the most intuitive training method according to Eq. (5), however, it does not work as well as *joint training* in terms of test negative log-likelihood (NLL), which is observed both in our experiments (Table B.6) and by Bauer & Mnih (2019).

- **Joint training** (Tomczak & Welling, 2018; Bauer & Mnih, 2019): $p_\lambda(\mathbf{z})$ is jointly trained along with $q_\phi(\mathbf{z}|\mathbf{x})$ and $p_\theta(\mathbf{x}|\mathbf{z})$, by directly maximizing Eq. (7).

- **Iterative training**: Proposed by us, we alternate *between* training $p_\theta(\mathbf{x}|\mathbf{z})$ & $q_\phi(\mathbf{z}|\mathbf{x})$ *and* training $p_\lambda(\mathbf{z})$, for multiple iterations. The first iteration to train $p_\theta(\mathbf{x}|\mathbf{z})$ & $q_\phi(\mathbf{z}|\mathbf{x})$ should use the unit Gaussian prior. Early-stopping should be performed during the whole process if necessary. See Algorithm B.1 for detailed procedure of this strategy. We adopt this method mainly for investigating why *post-hoc training* does not work as well as *joint training*.

To train $p_\lambda(\mathbf{z})$ with *post-hoc* strategy, we start from a trained VAE, adding RealNVP prior onto it, and optimizing the RealNVP $f_\lambda(\mathbf{z})$ for 3,200 epochs, with learning rate set to $10^{-3}$ in the first 1,600 epochs, and $10^{-4}$ in the final 1,600 epochs.

---

**Algorithm B.1** Pseudocode for iterative training.

**Iteration 1a**: Train $q_\phi(\mathbf{z}|\mathbf{x})$ and $p_\theta(\mathbf{x}|\mathbf{z})$, with $p_\lambda(\mathbf{z}) = \mathcal{N}(\mathbf{0}, \mathbf{I})$.
**Iteration 1b**: Train $p_\lambda(\mathbf{z})$ for $2M$ epochs, with fixed $q_\phi(\mathbf{z}|\mathbf{x})$ and $p_\theta(\mathbf{x}|\mathbf{z})$.
**for** i = 2 … I **do**
    **Iteration *i*a**: Train $q_\phi(\mathbf{z}|\mathbf{x})$ and $p_\theta(\mathbf{x}|\mathbf{z})$ for $M$ epochs, with fixed $p_\lambda(\mathbf{z})$.
    **Iteration *i*b**: Train $p_\lambda(\mathbf{z})$ for $M$ epochs, with fixed $q_\phi(\mathbf{z}|\mathbf{x})$ and $p_\theta(\mathbf{x}|\mathbf{z})$.
**end for**

---

Algorithm B.1 is the pseudocode of *iterative training* strategy. For Iteration 1a, all hyper-parameters are the same with training a standard VAE, where in particular, the number of training epochs is set to 2,400. For Iteration 1b, the learning rate is $10^{-3}$ for the first $M$ epochs, and is $10^{-4}$ for the next $M$ epochs. For all the next iterations, learning rate is always $10^{-4}$. The number of iterations $I$ is chosen to be 16, and the number of epochs $M$ is chosen to be 100, for MNIST, FashionMNIST and Omniglot. For StaticMNIST, we find it overfits after only a few iterations, thus we choose $I$ to be 4, and $M$ to be 400. With these hyper-parameters, $q_\phi(\mathbf{z}|\mathbf{x})$, $p_\theta(\mathbf{x}|\mathbf{z})$ and $p_\lambda(\mathbf{z})$ are *iteratively trained* for totally 3,200 epochs on all datasets (starting from Iteration 1b), after the pre-training step (Iteration 1a).

**Regularization term for $q_\phi(\mathbf{z})$**   In order to compare some metrics (*e.g.*, the *active units*) of VAE + RealNVP prior with those of standard VAE, we introduce an additional regularization term for $q_\phi(\mathbf{z})$, such that $q_\phi(\mathbf{z})$ of VAE using RealNVP prior would have roughly zero mean and unit variance, just as a standard VAE with unit Gaussian prior. The regularization term for $q_\phi(\mathbf{z})$ (denoted as $\text{Reg}\left[q_\phi(\mathbf{z})\right]$) and the final training objective augmented with the regularization term (denoted as $\widetilde{\mathcal{L}}(\mathbf{x}; \lambda, \theta, \phi)$) are:

$$\text{Reg}\left[q_\phi(\mathbf{z})\right] = \frac{1}{\text{Dim}(\mathbf{z})} \sum_{k=1}^{\text{Dim}(\mathbf{z})} \left[ \left(\text{Mean}(z_k)\right)^2 + \left(\text{Var}(z_k) - 1\right)^2 \right]$$

$$\widetilde{\mathcal{L}}(\lambda, \theta, \phi) = \mathbb{E}_{p^\star(\mathbf{x})} \mathcal{L}(\mathbf{x}; \lambda, \theta, \phi) + \text{Reg}\left[q_\phi(\mathbf{z})\right]$$

where $z_k$ is the $k$-th dimension of $\mathbf{z}$, $\mathrm{Dim}(\mathbf{z})$ is the number of dimensions, $\mathrm{Mean}(z_k) = \mathbb{E}_{p^\star(\mathbf{x})} \mathbb{E}_{q_\phi(\mathbf{z}|\mathbf{x})} [z_k]$ and $\mathrm{Var}(z_k) = \mathbb{E}_{p^\star(\mathbf{x})} \mathbb{E}_{q_\phi(\mathbf{z}|\mathbf{x})} \left[ (z_k - \mathrm{Mean}(z_k))^2 \right]$ are the mean and variance of each dimension.

Table B.1: Avg. $\mathrm{Mean}(z_k)$ and $\mathrm{Var}(z_k)$ of regularized/un-regularized ResnetVAE with RealNVP prior.

| Datasets | regularized | | un-regularized | |
|---|---|---|---|---|
| | Avg. $\mathrm{Mean}(z_k)$ | Avg. $\mathrm{Var}(z_k)$ | Avg. $\mathrm{Mean}(z_k)$ | Avg. $\mathrm{Var}(z_k)$ |
| StaticMNIST | $-0.02 \pm 0.03$ | $0.93 \pm 0.01$ | $0.05 \pm 0.37$ | $1.50 \pm 0.02$ |
| MNIST | $0.00 \pm 0.00$ | $0.98 \pm 0.01$ | $-0.01 \pm 0.02$ | $0.78 \pm 0.04$ |
| FashionMNIST | $0.00 \pm 0.00$ | $0.98 \pm 0.00$ | $0.03 \pm 0.03$ | $0.83 \pm 0.04$ |
| Omniglot | $0.00 \pm 0.02$ | $0.95 \pm 0.00$ | $0.00 \pm 0.01$ | $0.24 \pm 0.01$ |

Table B.1 shows the average $\mathrm{Mean}(z_k)$ and $\mathrm{Var}(z_k)$ of ResnetVAE with RealNVP prior, computed on test data. Average $\mathrm{Mean}(z_k)$ is defined as $\frac{1}{\mathrm{Dim}(\mathbf{z})} \sum_{k=1}^{\mathrm{Dim}(\mathbf{z})} \mathrm{Mean}(z_k)$, while average $\mathrm{Var}(z_k)$ is defined as $\frac{1}{\mathrm{Dim}(\mathbf{z})} \sum_{k=1}^{\mathrm{Dim}(\mathbf{z})} \mathrm{Var}(z_k)$. The means and standard deviations of the above table are computed *w.r.t.* repeated experiments. Using the regularization term for $q_\phi(\mathbf{z})$ makes the $\mathrm{Mean}(z_k)$ and $\mathrm{Var}(z_k)$ of $\mathbf{z}$ samples close to $\mathcal{N}(\mathbf{0}, \mathbf{I})$, which is in sharp contrast with the un-regularized case. For fair comparison in Table 5 and Fig. 3, it is crucial to have $\mathrm{Mean}(z_k)$ and $\mathrm{Var}(z_k)$ close to $\mathcal{N}(\mathbf{0}, \mathbf{I})$. Test NLLs are not reported here, because we find no significant difference between regularized and un-regularized models in terms of test NLLs.

**Evaluation** The *negative log-likelihood* (NLL), the *reconstruction loss* and the KL divergence $\mathbb{E}_{p^\star(\mathbf{x})} D_{\mathrm{KL}}(q_\phi(\mathbf{z}) \| p_\lambda(\mathbf{z}))$ are estimated with 1,000 $\mathbf{z}$ samples from $q_\phi(\mathbf{z}|\mathbf{x}^{(i)})$ for each $\mathbf{x}^{(i)}$ from test data:

$$\mathrm{NLL} \approx \frac{1}{N} \sum_{i=1}^{N} \mathrm{LogMeanExp}_{j=1}^{1000} \left[ \log p_\theta(\mathbf{x}^{(i)}|\mathbf{z}^{(i,j)}) + \log p_\lambda(\mathbf{z}^{(i,j)}) - \log q_\phi(\mathbf{z}^{(i,j)}|\mathbf{x}^{(i)}) \right]$$

$$\mathrm{Reconstruction\ Loss} \approx \frac{1}{1000N} \sum_{i=1}^{N} \sum_{j=1}^{1000} \left[ \log p_\theta(\mathbf{x}^{(i)}|\mathbf{z}^{(i,j)}) \right]$$

$$\mathbb{E}_{p^\star(\mathbf{x})} D_{\mathrm{KL}}(q_\phi(\mathbf{z}) \| p_\lambda(\mathbf{z})) \approx \frac{1}{1000N} \sum_{i=1}^{N} \sum_{j=1}^{1000} \left[ \log q_\phi(\mathbf{z}^{(i,j)}|\mathbf{x}^{(i)}) - \log p_\lambda(\mathbf{z}^{(i,j)}) \right]$$

where each $\mathbf{z}^{(i,j)}$ is one sample from $q_\phi(\mathbf{z}|\mathbf{x}^{(i)})$.

$\mathrm{LogMeanExp}_{j=1}^{L} \left[ f(\mathbf{x}^{(i)}, \mathbf{z}^{(i,j)}) \right]$ is defined as:

$$\mathrm{LogMeanExp}_{j=1}^{L} \left[ f(\mathbf{x}^{(i)}, \mathbf{z}^{(i,j)}) \right] = f_{max} + \log \frac{1}{L} \sum_{j=1}^{L} \left[ \exp \left( f(\mathbf{x}^{(i)}, \mathbf{z}^{(i,j)}) - f_{max} \right) \right]$$

where $f_{max} = \max_j f(\mathbf{x}^{(i)}, \mathbf{z}^{(i,j)})$.

We use 1,000 samples, because the two major previous works of us (*i.e.*, Tomczak & Welling (2018); Bauer & Mnih (2019)) both used this number of samples. It is true that some previous works may have used substantially larger number of samples (*e.g.*, Chen et al. (2017) used 4,096 samples), and according to (Grosse et al., 2015), the Monte Carlo estimator for NLL is a *stochastic lower-bound*, such that larger number of samples typically would lead to lower (yet better) estimated NLLs. However, in order to have fair comparison with (Tomczak & Welling, 2018; Bauer & Mnih, 2019), we decide to stick to 1,000 samples.

**Active units** *active units* (Burda et al., 2016) is defined as the number of latent dimensions whose variance is larger than 0.01. The variance of the $k$-th dimension is formulated as:

$$\mathrm{Var}_k = \mathrm{Var}_{p^\star(\mathbf{x})} \left( \mathbb{E}_{q_\phi(\mathbf{z}|\mathbf{x})} [z_k] \right)$$

where $z_k$ is the $k$-th dimension of $\mathbf{z}$. We compute $\text{Var}_k$ on the training data, while the inner expectation $\mathbb{E}_{q_\phi(\mathbf{z}|\mathbf{x})}[z_k]$ is estimated by drawing 1,000 samples of $\mathbf{z}$ for each $\mathbf{x}$.

## B.4 FORMULATION OF CLOSEST PAIRS OF $q_\phi(\mathbf{z}|\mathbf{x})$ AND OTHERS

**Closest pairs of** $q_\phi(\mathbf{z}|\mathbf{x})$  For each $\mathbf{x}^{(i)}$ from training data, we find the training point $\mathbf{x}^{(j)}$, whose posterior $q_\phi(\mathbf{z}|\mathbf{x}^{(j)})$ is the closest neighbor to $q_\phi(\mathbf{z}|\mathbf{x}^{(i)})$:

$$j = \arg\min_{j \neq i} \|\boldsymbol{\mu}_\phi(\mathbf{x}^{(j)}) - \boldsymbol{\mu}_\phi(\mathbf{x}^{(i)})\|$$

where $\boldsymbol{\mu}_\phi(\mathbf{x})$ is the mean of $q_\phi(\mathbf{z}|\mathbf{x})$, and $\|\cdot\|$ is the L2 norm. These two posteriors $q_\phi(\mathbf{z}|\mathbf{x}^{(i)})$ and $q_\phi(\mathbf{z}|\mathbf{x}^{(j)})$ are called a closest pair of $q_\phi(\mathbf{z}|\mathbf{x})$.

**Distance of a pair of** $q_\phi(\mathbf{z}|\mathbf{x})$  The distance $d_{ij}$ of a closest pair $q_\phi(\mathbf{z}|\mathbf{x}^{(i)})$ and $q_\phi(\mathbf{z}|\mathbf{x}^{(j)})$ is:

$$\mathbf{d}_{ij} = \boldsymbol{\mu}_\phi(\mathbf{x}^{(j)}) - \boldsymbol{\mu}_\phi(\mathbf{x}^{(i)})$$
$$d_{ij} = \|\mathbf{d}_{ij}\|$$

**Normalized distance of a pair of** $q_\phi(\mathbf{z}|\mathbf{x})$  For each closest pair $q_\phi(\mathbf{z}|\mathbf{x}^{(i)})$ and $q_\phi(\mathbf{z}|\mathbf{x}^{(j)})$, we compute its *normalized distance* $\widetilde{d_{ij}}$ by:

$$\widetilde{d_{ij}} = \frac{2d_{ij}}{\text{Std}[i; j] + \text{Std}[j; i]} \tag{B.3}$$

$\text{Std}[i; j]$ is formulated as:

$$\text{Std}[i; j] = \sqrt{\text{Var}_{q_\phi(\mathbf{z}|\mathbf{x}^{(i)})}\left( \left(\mathbf{z} - \boldsymbol{\mu}_\phi(\mathbf{x}^{(i)})\right) \cdot \frac{\mathbf{d}_{ij}}{d_{ij}} \right)} \tag{B.4}$$

where $\text{Var}_{q_\phi(\mathbf{z}|\mathbf{x}^{(i)})}(f(\mathbf{z}))$ is the variance of $f(\mathbf{z})$ *w.r.t.* $q_\phi(\mathbf{z}|\mathbf{x}^{(i)})$. We use 1,000 samples to estimate each $\text{Std}[i; j]$. Roughly speaking, the *normalized distance* $\widetilde{d_{ij}}$ can be viewed as "distance/std" along the direction of $\mathbf{d}_{ij}$, which indicates the scale of the "hole" between $q_\phi(\mathbf{z}|\mathbf{x}^{(i)})$ and $q_\phi(\mathbf{z}|\mathbf{x}^{(j)})$.

## B.5 EXPERIMENT RESULTS ON CIFAR10

We conduct experiments on Cifar10 (Krizhevsky, 2009). Although the results on Cifar10 are not as good as the state-of-the-art deep hierarchical VAEs, using RealNVP prior also shows great improvement on test NLLs and sampling quality over Gaussian prior, with just one latent variable.

**Architectures of ResnetVAE for Cifar10**  The latent variable $\mathbf{z}$ is a 3-d tensor, with shape $8 \times 8 \times 32$. The Gaussian posterior $q_\phi(\mathbf{z}|\mathbf{x})$ is derived as:

$$q_\phi(\mathbf{z}|\mathbf{x}) = \mathcal{N}(\boldsymbol{\mu}_\phi(h_\phi(\mathbf{x})), \boldsymbol{\sigma}_\phi^2(h_\phi(\mathbf{x}))\,\mathbf{I})$$
$$\boldsymbol{\mu}_\phi(h_\phi(\mathbf{x})) = h_\phi(\mathbf{x}) \to \text{LinearConv}_{1\times1}[8 \times 8 \times 32]$$
$$\log \boldsymbol{\sigma}_\phi(h_\phi(\mathbf{x})) = h_\phi(\mathbf{x}) \to \text{LinearConv}_{1\times1}[8 \times 8 \times 32]$$
$$h_\phi(\mathbf{x}) = \mathbf{x} \to \text{Resnet}[32 \times 32 \times 96 \to 32 \times 32 \times 96 \to 32 \times 32 \times 96$$
$$\to 16 \times 16 \times 192 \to 16 \times 16 \times 192 \to 16 \times 16 \times 192$$
$$\to 8 \times 8 \times 256 \to 8 \times 8 \times 256]$$

The generative distribution $p_\theta(\mathbf{x}|\mathbf{z})$ is derived as:

$$p_\theta(\mathbf{x}|\mathbf{z}) = \text{DiscretizedLogistic}\,(\boldsymbol{\mu}_\theta(h_\theta(\mathbf{z})), \mathbf{s}_\theta(h_\theta(\mathbf{z})))$$
$$\boldsymbol{\mu}_\theta(h_\theta(\mathbf{z})) = h_\theta(\mathbf{z}) \to \text{LinearConv}_{1\times1}[32 \times 32 \times 3]$$
$$\log \mathbf{s}_\theta(h_\theta(\mathbf{z})) = h_\theta(\mathbf{z}) \to \text{LinearConv}_{1\times1}[32 \times 32 \times 3]$$
$$h_\theta(\mathbf{z}) = \mathbf{z} \to \text{DeResnet}[8 \times 8 \times 256 \to 8 \times 8 \times 256$$
$$\to 16 \times 16 \times 192 \to 16 \times 16 \times 192 \to 16 \times 16 \times 192$$
$$\to 32 \times 32 \times 96 \to 32 \times 32 \times 96]$$

where DiscretizedLogistic is the discretized logistic distribution proposed by Kingma et al. (2016). Note we also clip $\log \mathbf{s}_\theta(h_\theta(\mathbf{z}))$ by $-11$, just as the for the log-std of $q_\phi(\mathbf{z}|\mathbf{x})$.

Table B.2: Test bpd on Cifar10.

| Model | bpd |
|---|---|
| *Models without PixelCNN decoder* | |
| VAE + IAF (Kingma et al., 2016) | 3.11 |
| BIVA (Maaløe et al., 2019) | **3.08** |
| **ResnetVAE** | 3.98 |
| **ResnetVAE + RealNVP $p(\mathbf{z})$**, $K = 50$ | 3.54 |
| *Models with PixelCNN decoder* | |
| VLAE (Chen et al., 2017) | 2.95 |

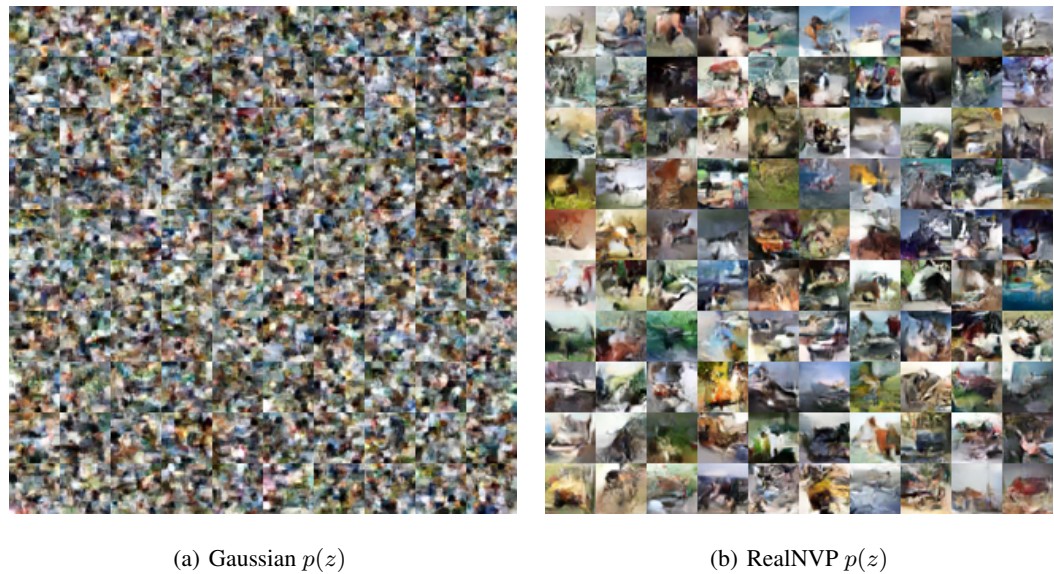

(a) Gaussian $p(z)$          (b) RealNVP $p(z)$

Figure B.1: Samples of: (a) ResnetVAE with Gaussian $p(z)$; and (b) ResnetVAE with RealNVP $p(z)$. Both models use just one latent variable of shape $8 \times 8 \times 32$.

**RealNVP prior** The RealNVP prior for Cifar10 consists of 50 blocks, while each block consists of an *invertible 1x1 conv* (Kingma & Dhariwal, 2018), a *coupling layer*, and an *actnorm*, which can be formulated as:

$$f_\lambda(\mathbf{z}) = \mathbf{z} \rightarrow f_1(\mathbf{h}_1) \rightarrow \cdots \rightarrow f_K(\mathbf{h}_K)$$

$$f_k(\mathbf{h}_k) = \mathbf{h}_k \rightarrow \text{InvertibleConv}_{1\times1} \rightarrow \text{CouplingLayer} \rightarrow \text{ActNorm}$$

$$\text{CouplingLayer}(\mathbf{u}) = \text{Concat}\big[\mathbf{u}_l, \mathbf{u}_r \odot \text{Sigmoid}\left(s_\lambda(h_{\lambda,k}(\mathbf{u}_l))\right) + t_\lambda(h_{\lambda,k}(\mathbf{u}_l))\big]$$

$$h_{\lambda,k}(\mathbf{u}_l) = \mathbf{u}_l \rightarrow \text{Resnet}[8 \times 8 \times 64 \rightarrow 8 \times 8 \times 64]$$

$$s_\lambda(h_{\lambda,k}(\mathbf{u}_l)) = h_{\lambda,k}(\mathbf{u}_l) \rightarrow \text{LinearConv}_{1\times1}[8 \times 8 \times 32]$$

$$t_\lambda(h_{\lambda,k}(\mathbf{u}_l)) = h_{\lambda,k}(\mathbf{u}_l) \rightarrow \text{LinearConv}_{1\times1}[8 \times 8 \times 32]$$

where $\mathbf{u}_l = \mathbf{u}_{:,:,0:\lfloor \text{Dim}(\mathbf{u})/2 \rfloor}$ is the left half channels of $\mathbf{u}$, and $\mathbf{u}_r = \mathbf{u}_{:,:,\lfloor \text{Dim}(\mathbf{u})/2 \rfloor:\text{Dim}(\mathbf{u})}$ is the right half channels.

**Experiment results** Table B.2 shows the test bpd of our ResnetVAE on Cifar10. Although the test bpd of our model is still far from those of deep hierarchical models, it is worth mention that BIVA used 15 latent variables to reach the state-of-the-art. Our experiments show that, adding a deep RealNVP prior can make the test bpd much better, even with just one latent variable. This suggests that a complicated prior can be very effective for reaching good test bpd, and is potentially helpful for reducing the depth of hierarchical VAEs.

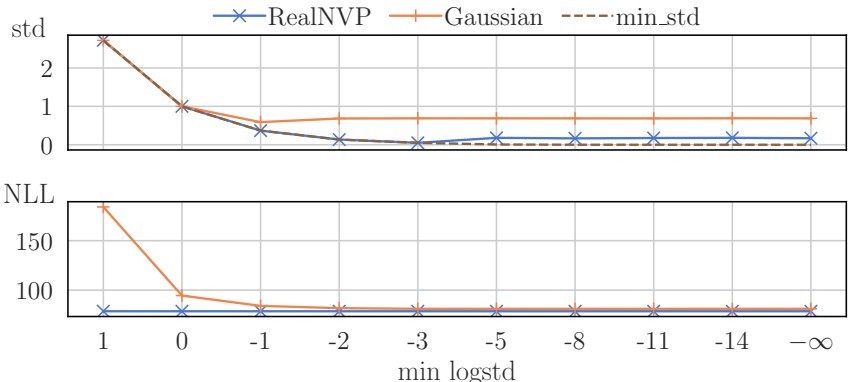

Figure B.2: The average learned std of Gaussian $q_\phi(\mathbf{z}|\mathbf{x})$ and the test NLL of ResnetVAE with Real-NVP and Gaussian $p(z)$ on MNIST. We also plot "min_std" as a reference, the minimum attainable std clipped by the configured mininum logstd.

The most surprising fact is that, in our experiments, it is not possible to obtain any meaningful samples using ResnetVAE with Gaussian $p(z)$. In contrary, the samples obtained using ResnetVAE with RealNVP $p(z)$ is much more like natural images.

### B.6    IMPACT OF THE MINIMUM CLIPPING VALUE FOR THE STD OF $q_\phi(\mathbf{z}|\mathbf{x})$

We have mentioned in Section 5.1 that we clip the std of Gaussian posterior $q_\phi(\mathbf{z}|\mathbf{x})$ by a minimum value of $e^{-11} \approx 1.67 \times 10^{-5}$, so as to avoid unstable training. The clipping is actually done by directly clipping the linear activation of log-std of $q_\phi(\mathbf{z}|\mathbf{x})$ by $-11$, and then obtain std, instead of clipping the std itself. Formally, the clipped log-std and the std are derived by:

$$\text{logstd} = \text{maximum}\left\{\sigma_\phi(f_\phi(\mathbf{z})), -11\right\}$$
$$\text{std} = \exp\left(\text{logstd}\right)$$

where $f_\phi(\mathbf{z})$ is the hidden feature network for $q_\phi(\mathbf{z}|\mathbf{x})$, and $\sigma_\phi(\cdot)$ is a linear fully-connected layer. We use the clipped log-std directly instead of the std whenever possible in computing the log-likelihood of Gaussian $q_\phi(\mathbf{z}|\mathbf{x})$, which can be formulated as:

$$\log q_\phi(\mathbf{z}|\mathbf{x}) = -\frac{1}{2}\log(2\pi) - \text{logstd} - \frac{1}{2}\exp\left(-2\text{logstd}\right) \cdot \left(\mathbf{x} - \mu_\phi(f_\phi(\mathbf{z}))\right)^2$$

where $\mu_\phi(\mathbf{z})$ is a linear fully-connected layer, producing the mean of $q_\phi(\mathbf{z}|\mathbf{x})$. Std clipping is widely used in many open-source implementations of VAEs. However, this trick is seldom discussed in published papers, which might suggest more theoretical work should be done on this topic.

The clipping value $-11$ in our experiments is not chosen for better model performance. The term $\exp\left(-2\text{logstd}\right)$ will have a maximum value of approximately $3.58 \times 10^9$ with this value, which we think is safe enough for not having NaNs in training, considering that we mainly use float32 in our neural network implementation.

In fact, we find that the test NLL of the model and the learned std of $q_\phi(\mathbf{z}|\mathbf{x})$ is insensitive to the clipping value under a certain threshold, which is demonstrated in Fig. B.2 and Table B.3. As long as the minimum std is small enough, both the RealNVP prior and the Gaussian prior could induce adaptive learned std of Gaussian posterior $q_\phi(\mathbf{z}|\mathbf{x})$. One surprising fact is that, there is a turning point where the clipping value is not small enough yet (*i.e.*, min logstd being $-3$ for RealNVP $p(z)$, and $-1$ for Gaussian $p(z)$), the final learned std is smaller than the learned std with smaller clipping values. The cause of this phenomenon is not analyzed in our paper, and we leave it as a future work.

Table B.3: The minimum std ("min_std"), average learned std of Gaussian $q_\phi(\mathbf{z}|\mathbf{x})$ ("std"), the ratio of minimum std with respect to the learned std ("min_std / std"), the test NLL and the active units ("au") of ResnetVAE with RealNVP $p(z)$ and Gaussian $p(z)$ on MNIST.

| ResnetVAE with | min logstd | min_std | std | min_std / std | NLL | au |
|---|---|---|---|---|---|---|
| RealNVP $p(z)$ | 1 | 2.7183 | 2.7183 | 1.0000 | 78.61 | 40 |
| | 0 | 1.0000 | 1.0000 | 1.0000 | 78.57 | 40 |
| | -1 | 0.3679 | 0.3679 | 1.0000 | 78.58 | 40 |
| | -2 | 0.1353 | 0.1353 | 1.0000 | 78.57 | 40 |
| | -3 | 0.0498 | 0.0500 | 0.9967 | 78.59 | 37 |
| | -5 | 0.0067 | 0.1796 | 0.0375 | 78.56 | 40 |
| | -8 | 0.0003 | 0.1662 | 0.0020 | 78.56 | 40 |
| | -11 | 0.0000 | 0.1750 | 0.0001 | 78.60 | 40 |
| | -14 | 0.0000 | 0.1791 | 0.0000 | 78.56 | 40 |
| | $-\infty$ | 0.0000 | 0.1693 | 0.0000 | 78.58 | 40 |
| Normal | 1 | 2.7183 | 2.7183 | 1.0000 | 184.16 | 40 |
| | 0 | 1.0000 | 1.0000 | 1.0000 | 94.46 | 40 |
| | -1 | 0.3679 | 0.5890 | 0.6246 | 84.03 | 36 |
| | -2 | 0.1353 | 0.6839 | 0.1979 | 81.75 | 25 |
| | -3 | 0.0498 | 0.6877 | 0.0724 | 81.07 | 25 |
| | -5 | 0.0067 | 0.6882 | 0.0098 | 81.05 | 26 |
| | -8 | 0.0003 | 0.6871 | 0.0005 | 81.04 | 25 |
| | -11 | 0.0000 | 0.6860 | 0.0000 | 81.03 | 26 |
| | -14 | 0.0000 | 0.6896 | 0.0000 | 81.06 | 25 |
| | $-\infty$ | 0.0000 | 0.6873 | 0.0000 | 81.08 | 26 |

### B.7 COMPARISON OF OUR WORK AND THE PREVIOUS PAPER USING REALNVP PRIOR

The hyper-parameters of Huang et al. (2017) are quite different from ours: they only used 3 layers of ResNet blocks in their $q_\phi(\mathbf{z}|\mathbf{x})$ and 4 layers in their $p_\theta(\mathbf{x}|\mathbf{z})$. Also, they trained their ResConv VAE on StaticMNIST only for 300 epochs, using an initial learning rate of 0.0005 without learning rate annealing, and without validation set. In addition, we adopt invertible dense and actnorm in our RealNVP flow, while they just used the original RealNVP flow. Most parts of our ResnetVAE with RealNVP prior have larger complexity than the ResConv VAE of Huang et al. (2017), expect that they used 2 hidden layers in each coupling layer, while we just used 1.

Table B.4: Test NLL of our ResnetVAE with low complexity and ResConv VAE (Huang et al., 2017) on StaticMNIST. "pure" refers to the original RealNVP flow, without invertible dense and actnorm.

| Model | $p(z)$ flow | $q(z\|x)$ flow | Epochs | Initial lr | Early stopping | NLL |
|---|---|---|---|---|---|---|
| Our ResnetVAE | pure, $K = 12$ | / | 300 | 0.0005 | no | 81.05 |
| Low Complexity | pure, $K = 8$ | pure, $K = 8$ | 300 | 0.0005 | no | 80.88 |
| | $K = 12$ | / | 300 | 0.0005 | no | 80.67 |
| | $K = 8$ | $K = 8$ | 300 | 0.0005 | no | 80.66 |
| ResConv VAE | pure, $K = 12$ | / | 300 | 0.0005 | no | 81.44 |
| (Huang et al., 2017) | pure, $K = 8$ | pure, $K = 8$ | 300 | 0.0005 | no | 80.81 |

By removing the invertible dense and actnorm components from the RealNVP prior, enforcing the encoder and the decoder to have similar model complexity, using the same number of training epochs and learning rate, and not using validation set to do early-stopping, we obtained similar results as Huang et al. (2017) (the first two rows of "Ours"). Adding invertible dense and actnorm (the third and fourth rows of "Ours") will improve the test NLL, suggesting that the invertible dense and actnorm can indeed bring benefit to the model performance.

According to the above results, it is reasonable to believe that, it is the deeper RealNVP prior, more complicated model architecture, and longer training time, that give us better test NLL than Huang

et al. (2017). In conclusion, there may not be too many differences between our model and that of Huang et al. (2017). However, we tune the hyper-parameters much better, and demonstrate that ResnetVAE with RealNVP prior and just one latent variable can work as well as many deep hierarchical VAEs on MNIST-like datasets. The additional analysis should also be our novel contribution, when compared to the work of Huang et al. (2017).

## B.8    DISCUSSIONS ABOUT $D_{\mathrm{KL}}(q_\phi(\mathbf{z})\|p_\lambda(\mathbf{z}))$

The KL divergence between the *aggregated posterior* and the prior, *i.e.*, $D_{\mathrm{KL}}(q_\phi(\mathbf{z})\|p_\lambda(\mathbf{z}))$, is first analyzed by Hoffman & Johnson (2016) as one component of the ELBO decomposition (5). Since $q_\phi(\mathbf{z}) = \int q_\phi(\mathbf{z}|\mathbf{x})\, p^\star(\mathbf{x})\, \mathrm{d}\mathbf{x}$ and $p_\lambda(\mathbf{z}) = \int p_\theta(\mathbf{z}|\mathbf{x})\, p_\theta(\mathbf{x})\, \mathrm{d}\mathbf{x}$, Rosca et al. (2018) used this KL divergence as a metric to quantify the approximation quality of both $p_\theta(\mathbf{x})$ to $p^\star(\mathbf{x})$, and $q_\phi(\mathbf{z}|\mathbf{x})$ to $p_\theta(\mathbf{z}|\mathbf{x})$. As we are focusing on learning the prior, $D_{\mathrm{KL}}(q_\phi(\mathbf{z})\|p_\lambda(\mathbf{z}))$ can in turn be a metric for quantifying whether $p_\lambda(\mathbf{z})$ is close enough to the *aggregated posterior* $q_\phi(\mathbf{z})$.

The evaluation of $D_{\mathrm{KL}}(q_\phi(\mathbf{z})\|p_\lambda(\mathbf{z}))$, however, is not an easy task. One way to estimate the KL divergence of two arbitrary distributions is the density ratio trick (Sugiyama et al., 2012; Mescheder et al., 2017), where $D_{\mathrm{KL}}(q_\phi(\mathbf{z})\|p_\lambda(\mathbf{z}))$ is estimated by a separately trained neural network classifier. However, Rosca et al. (2018) revealed that such approach can under-estimate $D_{\mathrm{KL}}(q_\phi(\mathbf{z})\|p_\lambda(\mathbf{z}))$, and the training may even diverge when $\mathbf{z}$ has hundreds of or more dimensions.

Another approach is to directly estimate $D_{\mathrm{KL}}(q_\phi(\mathbf{z})\|p_\lambda(\mathbf{z}))$ by Monte Carlo integration. The KL divergence can be rewritten into the following form:

$$
\begin{aligned}
D_{\mathrm{KL}}(q_\phi(\mathbf{z})\|p_\lambda(\mathbf{z})) &= \int q_\phi(\mathbf{z}) \log \frac{q_\phi(\mathbf{z})}{p_\lambda(\mathbf{z})}\, \mathrm{d}\mathbf{z} \\
&= \int \left( \int q_\phi(\mathbf{z}|\mathbf{x})\, p^\star(\mathbf{x})\, \mathrm{d}\mathbf{x} \right) \log \frac{\int q_\phi(\mathbf{z}|\mathbf{x}')\, p^\star(\mathbf{x}')\, \mathrm{d}\mathbf{x}'}{p_\lambda(\mathbf{z})}\, \mathrm{d}\mathbf{z} \\
&= \int p^\star(\mathbf{x}) \int q_\phi(\mathbf{z}|\mathbf{x}) \log \frac{\int q_\phi(\mathbf{z}|\mathbf{x}')\, p^\star(\mathbf{x}')\, \mathrm{d}\mathbf{x}'}{p_\lambda(\mathbf{z})}\, \mathrm{d}\mathbf{z}\, \mathrm{d}\mathbf{x} \\
&= \mathbb{E}_{p^\star(\mathbf{x})}\, \mathbb{E}_{q_\phi(\mathbf{z}|\mathbf{x})} \left[ \log \mathbb{E}_{p^\star(\mathbf{x}')} \left[ q_\phi(\mathbf{z}|\mathbf{x}') \right] - \log p_\lambda(\mathbf{z}) \right]
\end{aligned}
\tag{B.5}
$$

Rosca et al. (2018) has already proposed a Monte Carlo based algorithm to estimate $D_{\mathrm{KL}}(q_\phi(\mathbf{z})\|p_\lambda(\mathbf{z}))$, in case that the training data is statically binarized. Since we mainly use dynamically binarized datasets, we slightly modified the algorithm according to Eq. (B.5), to allow sampling multiple $\mathbf{x}$ from each image. Our algorithm is:

Surprisingly, we find that increasing $n_x$ will cause the estimated $D_{\mathrm{KL}}(q_\phi(\mathbf{z})\|p_\lambda(\mathbf{z}))$ to decrease on MNIST, see Table B.5. Given that our $n_z = 10$ for all $n_x$, the number of our sampled $\mathbf{z}$ (even when $n_x = 1$) is $10 \times 60,000 = 6 \times 10^5$, which should not be too small, since Rosca et al. (2018) only used $10^6$ $\mathbf{z}$ in their experiments. When $n_x = 8$, the number of $\mathbf{z}$ is $8 \times 10 \times 60,000 = 4.8 \times 10^6$, which is 4.8x larger than Rosca et al. (2018), not to mention the inner expectation $\mathbb{E}_{p^\star(\mathbf{x}')}[q_\phi(\mathbf{z}|\mathbf{x}')]$ is estimated with 8x larger number of $\mathbf{x}'$. There should be in total 38.4x larger number of $\log q_\phi(\mathbf{z}|\mathbf{x}')$ computed for estimating $D_{\mathrm{KL}}(q_\phi(\mathbf{z})\|p_\lambda(\mathbf{z}))$ than Rosca et al. (2018), which has already costed about 3 days on 4 GTX 1080 Ti graphical cards. We believe such a large number of Monte Carlo samples should be sufficient for any algorithm with good behavior.

Table B.5: $D_{\mathrm{KL}}(q_\phi(\mathbf{z})\|p_\lambda(\mathbf{z}))$ of a ResnetVAE with RealNVP prior trained on MNIST, estimated by Algorithm B.2. $n_z = 10$ for all $n_x$. We only tried $n_x$ for up to 8, due to the growing computation time of $O(n_x^2)$.

| $n_x$ | 1 | 2 | 3 | 5 | 8 |
|---|---|---|---|---|---|
| $D_{\mathrm{KL}}(q_\phi(\mathbf{z})\|p_\lambda(\mathbf{z}))$ | 15.279 | 14.623 | 14.254 | 13.796 | 13.392 |

According to the above observation, we suspect there must be some flaw in Algorithm B.2. Because of this, we do not adopt Algorithm B.2 to estimate $D_{\mathrm{KL}}(q_\phi(\mathbf{z})\|p_\lambda(\mathbf{z}))$.

Since no mature method has been published to estimate $D_{\mathrm{KL}}(q_\phi(\mathbf{z})\|p_\lambda(\mathbf{z}))$, we decide not to use $D_{\mathrm{KL}}(q_\phi(\mathbf{z})\|p_\lambda(\mathbf{z}))$ to measure how our learned $p_\lambda(\mathbf{z})$ approximates the *aggregated posterior* $q_\phi(\mathbf{z})$.

**Algorithm B.2** Pseudocode for estimating $D_{\mathrm{KL}}(q_\phi(\mathbf{z}) \| p_\lambda(\mathbf{z}))$ (denoted as marginal_kl) on dynamically binarized dataset, where $\boldsymbol{\mu}$ is the pixel intensities of each original image.

---

x_samples = []
**for** $\boldsymbol{\mu}$ in training dataset **do**
    **for** $i = 1 \ldots n_x$ **do**
        sample $\mathbf{x}$ from $\mathrm{Bernoulli}(\boldsymbol{\mu})$
        append $\mathbf{x}$ to x_samples
    **end for**
**end for**
marginal_kl = 0
**for** $\mathbf{x}$ in x_samples **do**
    **for** $i = 1 \ldots n_z$ **do**
        sample $\mathbf{z}$ from $q_\phi(\mathbf{z}|\mathbf{x})$
        posterior_list = []
        **for** $\mathbf{x}'$ in x_samples **do**
            append $\log q_\phi(\mathbf{z}|\mathbf{x}')$ to posterior_list
        **end for**
        $\log q_\phi(\mathbf{z}) = \mathrm{LogMeanExp}(\text{posterior\_list})$
        marginal_kl = marginal_kl $+ \log q_\phi(\mathbf{z}) - \log p_\lambda(\mathbf{z})$
    **end for**
**end for**
marginal_kl = marginal_kl$/(\mathrm{len}(\text{x\_samples}) \times n_z)$

---

## B.9    ITERATIVE TRAINING CAN LEAD TO IMPROVED RECONSTRUCTION LOSS AND INCREASED ACTIVE UNITS

Table B.6 shows the NLLs of *iterative training* and *post-hoc training* with ResnetVAE (see also Appendix B.3). Although still not comparable to *joint training*, both methods can bring large improvement in NLLs over standard VAE. Also, *iterative training* even further outperforms *post-hoc training* by a large margin.

Table B.7 shows that, compared with *post-hoc training*, *iterative training* can lead to larger *reconstruction loss* and increased number of *active units*. Section 5.5 suggests that the learned prior can encourage better trade-off between *reconstruction loss* and the KL divergence, thus a well-trained prior is essential for obtaining good $q_\phi(\mathbf{z}|\mathbf{x})$ and $p_\theta(\mathbf{x}|\mathbf{z})$. However, the prior is in turn determined by $q_\phi(\mathbf{z}|\mathbf{x})$ and $p_\theta(\mathbf{x}|\mathbf{z})$, according to Eq. (5). Thus, it is important to alternate *between* training $p_\theta(\mathbf{x}|\mathbf{z})$ & $q_\phi(\mathbf{z}|\mathbf{x})$ *and* training $p_\lambda(\mathbf{z})$, until they catch up wtih each other and reach an equilibrium.

Considering these reasons, it is not surprising why *iterative training* can result in larger (yet better) *reconstruction loss* than *post-hoc training*. It is also clear why *joint training* works the best: because $p_\theta(\mathbf{x}|\mathbf{z})$ & $q_\phi(\mathbf{z}|\mathbf{x})$, and $p_\lambda(\mathbf{z})$, are trained to catch up with each other at every step, thus the training is more sufficient when using *joint training* than using the other two techniques.

Table B.6: Test NLL of ResnetVAE, with prior trained by: *joint* training, *iterative* training, *post-hoc* training, and standard VAE ("none") as reference. Flow depth $K = 20$.

| | **ResnetVAE** | | | |
|---|---|---|---|---|
| **Datasets** | joint | iterative | post-hoc | none |
| StaticMNIST | **79.99** | 80.63 | 80.86 | 82.95 |
| MNIST | **78.58** | 79.61 | 79.90 | 81.07 |
| FashionMNIST | **224.09** | 224.88 | 225.22 | 226.17 |
| Omniglot | **93.61** | 94.43 | 94.87 | 96.99 |

Table B.7: Average *reconstruction loss* ("*recons*"), $\mathbb{E}_{p^\star(\mathbf{x})} D_{\mathrm{KL}}(q_\phi(\mathbf{z}|\mathbf{x}) \| p_\lambda(\mathbf{z}))$ ("*kl*") and *active units* ("*au*") of ResnetVAE with *iteratively trained* and *post-hoc* trained RealNVP priors.

| | iterative | | | post-hoc | | |
|---|---|---|---|---|---|---|
| Datasets | *recons* | *kl* | *au* | *recons* | *kl* | *au* |
| StaticMNIST | **-58.0** | 26.4 | **38.7** | -60.1 | **25.3** | 30 |
| DynamicMNIST | **-57.2** | 25.1 | **40** | -58.7 | **24.7** | 25.3 |
| FashionMNIST | **-207.8** | 19.4 | **64** | -208.9 | **19.0** | 27 |
| Omniglot | **-63.3** | 37.9 | **64** | -67.0 | **35.8** | 59.3 |

## B.10 ADDITIONAL QUANTITATIVE RESULTS

Additional quantitative results are shown in Table B.8 to B.20.

Table B.8: Test NLL on StaticMNIST. "†" and "‡" has the same meaning as Table 1.

| Model | NLL |
|---|---|
| *Models without PixelCNN decoder* | |
| ConvHVAE + Lars prior[†] (Bauer & Mnih, 2019) | 81.70 |
| ConvHVAE + VampPrior[†] (Tomczak & Welling, 2018) | 81.09 |
| ResConv + RealNVP prior (Huang et al., 2017) | 81.44 |
| VAE + IAF[‡] (Kingma et al., 2016) | 79.88 |
| BIVA[‡] (Maaløe et al., 2019) | **78.59** |
| **ConvVAE + RNVP** $p(\mathbf{z})$, $K = 50$ | $80.09 \pm 0.01$ |
| **ResnetVAE + RNVP** $p(\mathbf{z})$, $K = 50$ | $79.84 \pm 0.04$ |
| *Models with PixelCNN decoder* | |
| VLAE[‡] (Chen et al., 2017) | 79.03 |
| PixelHVAE + VampPrior[†] (Tomczak & Welling, 2018) | 79.78 |
| **PixelVAE + RNVP** $p(\mathbf{z})$, $K = 50$ | $\mathbf{79.01 \pm 0.03}$ |

Table B.9: Test NLL on MNIST. "†" and "‡" has the same meaning as Table 1.

| Model | NLL |
|---|---|
| *Models without PixelCNN decoder* | |
| ConvHVAE + Lars prior[†] (Bauer & Mnih, 2019) | 80.30 |
| ConvHVAE + VampPrior[†] (Tomczak & Welling, 2018) | 79.75 |
| VAE + IAF[‡] (Kingma et al., 2016) | $79.10 \pm 0.07$ |
| BIVA[‡] (Maaløe et al., 2019) | **78.41** |
| **ConvVAE + RNVP $p(\mathbf{z})$, $K = 50$** | $78.61 \pm 0.01$ |
| **ResnetVAE + RNVP $p(\mathbf{z})$, $K = 50$** | $78.49 \pm 0.01$ |
| *Models with PixelCNN decoder* | |
| VLAE[‡] (Chen et al., 2017) | 78.53 |
| PixelVAE[†] (Gulrajani et al., 2017) | 79.02 |
| PixelHVAE + VampPrior[†] (Tomczak & Welling, 2018) | 78.45 |
| **PixelVAE + RNVP $p(\mathbf{z})$, $K = 50$** | $\mathbf{78.12 \pm 0.04}$ |

Table B.10: Test NLL on Omniglot. "†" and "‡" has the same meaning as Table 1.

| Model | NLL |
|---|---|
| *Models without PixelCNN decoder* | |
| ConvHVAE + Lars prior[†] (Bauer & Mnih, 2019) | 97.08 |
| ConvHVAE + VampPrior[†] (Tomczak & Welling, 2018) | 97.56 |
| BIVA[‡] (Maaløe et al., 2019) | **91.34** |
| **ConvVAE + RNVP $p(\mathbf{z})$, $K = 50$** | $93.62 \pm 0.02$ |
| **ResnetVAE + RNVP $p(\mathbf{z})$, $K = 50$** | $93.52 \pm 0.02$ |
| *Models with PixelCNN decoder* | |
| VLAE[‡] (Chen et al., 2017) | 89.83 |
| PixelHVAE + VampPrior[†] (Tomczak & Welling, 2018) | 89.76 |
| **PixelVAE + RNVP $p(\mathbf{z})$, $K = 50$** | $\mathbf{89.60 \pm 0.01}$ |

Table B.11: Test NLL on FashionMNIST. "†" and "‡" has the same meaning as Table 1.

| Model | NLL |
|---|---|
| *Models without PixelCNN decoder* | |
| ConvHVAE + Lars prior[†] (Bauer & Mnih, 2019) | 225.92 |
| **ConvVAE + RNVP $p(\mathbf{z})$, $K = 50$** | $224.64 \pm 0.01$ |
| **ResnetVAE + RNVP $p(\mathbf{z})$, $K = 50$** | $\mathbf{224.07 \pm 0.01}$ |
| *Models with PixelCNN decoder* | |
| **PixelVAE + RNVP $p(\mathbf{z})$, $K = 50$** | $\mathbf{223.36 \pm 0.06}$ |

## B.11 ADDITIONAL QUALITATIVE RESULTS

Additional qualitative results are shown in Fig. B.3 to B.5.

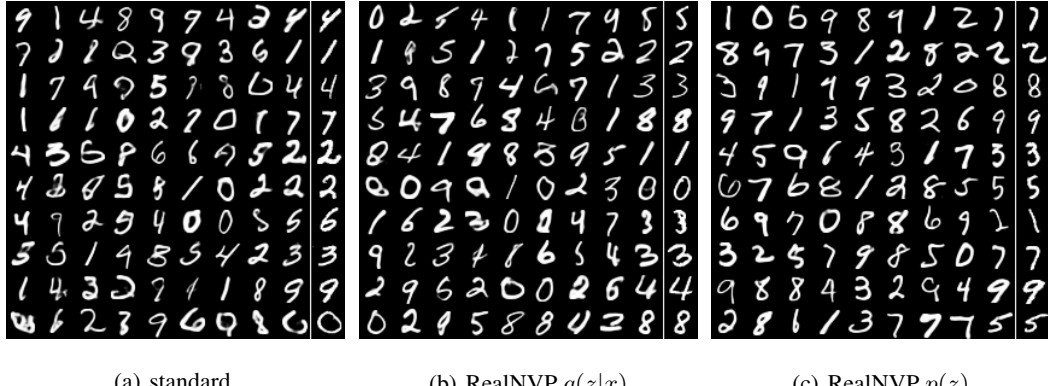

(a) standard      (b) RealNVP $q(z|x)$      (c) RealNVP $p(z)$

Figure B.3: Samples from ResnetVAE trained on MNIST. The last column of each 10x10 grid shows the training set images, most similar to the second-to-last column in pixel-wise L2 distance.

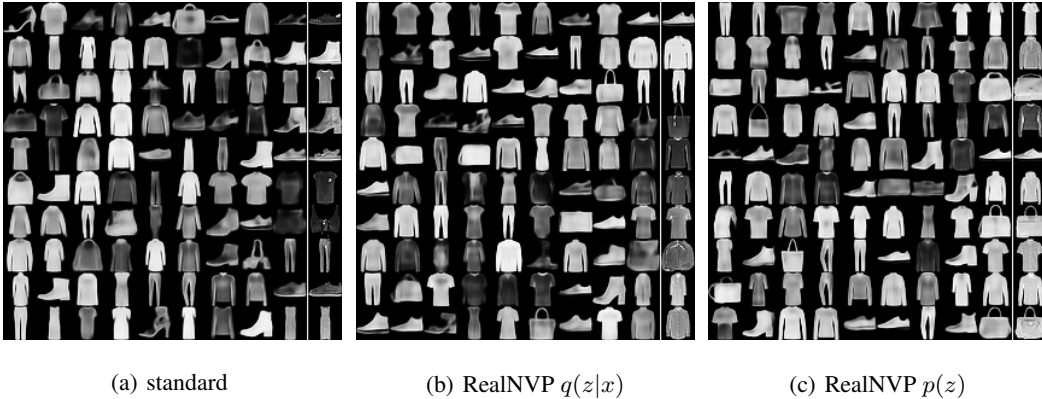

(a) standard      (b) RealNVP $q(z|x)$      (c) RealNVP $p(z)$

Figure B.4: Samples from ResnetVAE trained on FashionMNIST. The last column of each 10x10 grid shows the training set images, most similar to the second-to-last column in pixel-wise L2 distance.

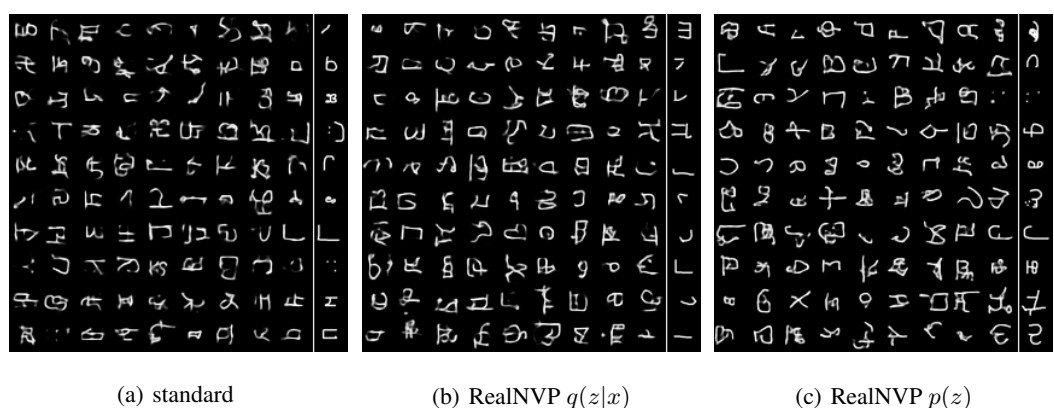

(a) standard      (b) RealNVP $q(z|x)$      (c) RealNVP $p(z)$

Figure B.5: Samples from ResnetVAE trained on Omniglot. The last column of each 10x10 grid shows the training set images, most similar to the second-to-last column in pixel-wise L2 distance.

## B.12 ADDITIONAL RESULTS: ABLATION STUDY AND POSTERIOR OVERLAPPING

Additional results about ablation study and posterior overlapping are shown in Table B.12 to B.19.

Table B.12: Average test NLL (lower is better) of different models, with Gaussian prior & Gaussian posterior ("standard"), Gaussian prior & RealNVP posterior ("RNVP $q(z|x)$"), and RealNVP prior & Gaussian posterior ("RNVP $p(z)$"). Flow depth $K = 20$.

| Models | | Datasets | | | |
|---|---|---|---|---|---|
| | | StaticMNIST | MNIST | FashionMNIST | Omniglot |
| DenseVAE | standard | $88.84 \pm 0.05$ | $84.48 \pm 0.03$ | $228.60 \pm 0.03$ | $106.42 \pm 0.14$ |
| | RNVP $q(z|x)$ | $86.07 \pm 0.11$ | $82.53 \pm 0.00$ | $227.79 \pm 0.01$ | $102.97 \pm 0.06$ |
| | RNVP $p(z)$ | $\mathbf{84.87 \pm 0.05}$ | $\mathbf{80.43 \pm 0.01}$ | $\mathbf{226.11 \pm 0.02}$ | $\mathbf{102.19 \pm 0.12}$ |
| ConvVAE | standard | $83.63 \pm 0.01$ | $82.14 \pm 0.01$ | $227.51 \pm 0.08$ | $97.87 \pm 0.02$ |
| | RNVP $q(z|x)$ | $81.11 \pm 0.03$ | $80.09 \pm 0.01$ | $226.03 \pm 0.00$ | $94.90 \pm 0.03$ |
| | RNVP $p(z)$ | $\mathbf{80.06 \pm 0.07}$ | $\mathbf{78.67 \pm 0.01}$ | $\mathbf{224.65 \pm 0.01}$ | $\mathbf{93.68 \pm 0.01}$ |
| ResnetVAE | standard | $82.95 \pm 0.09$ | $81.07 \pm 0.03$ | $226.17 \pm 0.05$ | $96.99 \pm 0.04$ |
| | RNVP $q(z|x)$ | $80.97 \pm 0.05$ | $79.53 \pm 0.03$ | $225.02 \pm 0.01$ | $94.30 \pm 0.02$ |
| | RNVP $p(z)$ | $\mathbf{79.99 \pm 0.02}$ | $\mathbf{78.58 \pm 0.01}$ | $\mathbf{224.09 \pm 0.01}$ | $\mathbf{93.61 \pm 0.04}$ |
| PixelVAE | standard | $79.47 \pm 0.02$ | $78.64 \pm 0.02$ | $224.22 \pm 0.06$ | $89.83 \pm 0.04$ |
| | RNVP $q(z|x)$ | $79.09 \pm 0.01$ | $78.41 \pm 0.01$ | $223.81 \pm 0.00$ | $89.69 \pm 0.01$ |
| | RNVP $p(z)$ | $\mathbf{78.92 \pm 0.02}$ | $\mathbf{78.15 \pm 0.04}$ | $\mathbf{223.40 \pm 0.07}$ | $\mathbf{89.61 \pm 0.03}$ |

Table B.13: Test NLL of ResnetVAE on MNIST, with RealNVP posterior ("$q(z|x)$"), RealNVP prior ("$p(z)$"), and RealNVP prior & posterior ("both"). Flow depth $K$ is $2K_0$ for the posterior or the prior in "$q(z|x)$" and "$p(z)$", while $K_0$ for both the posterior and the prior in "both".

| ResnetVAE with | $K_0$ | | | | |
|---|---|---|---|---|---|
| | 1 | 2 | 5 | 10 | 20 |
| $q(z|x)$, $K = 2K_0$ | $80.29 \pm 0.03$ | $80.00 \pm 0.00$ | $79.68 \pm 0.01$ | $79.53 \pm 0.02$ | $79.49 \pm 0.02$ |
| both, $K = K_0$ | $79.85 \pm 0.01$ | $79.42 \pm 0.02$ | $79.01 \pm 0.02$ | $78.71 \pm 0.00$ | $78.56 \pm 0.01$ |
| $p(z)$, $K = 2K_0$ | $\mathbf{79.58 \pm 0.02}$ | $\mathbf{79.21 \pm 0.01}$ | $\mathbf{78.75 \pm 0.01}$ | $\mathbf{78.58 \pm 0.01}$ | $\mathbf{78.51 \pm 0.02}$ |

Table B.14: Test reconstruction loss of ResnetVAE on MNIST, with RealNVP posterior ("$q(z|x)$"), RealNVP prior ("$p(z)$"), and RealNVP prior & posterior ("both"). Flow depth $K$ is $2K_0$ for the posterior or the prior in "$q(z|x)$" and "$p(z)$", while $K_0$ for both the posterior and the prior in "both".

| ResnetVAE with | $K_0$ | | | | |
|---|---|---|---|---|---|
| | 1 | 2 | 5 | 10 | 20 |
| $q(z|x)$, $K = 2K_0$ | $-57.20 \pm 0.05$ | $-56.82 \pm 0.01$ | $-56.37 \pm 0.04$ | $-56.15 \pm 0.05$ | $-55.98 \pm 0.05$ |
| both, $K = K_0$ | $-56.34 \pm 0.01$ | $-55.64 \pm 0.02$ | $-54.68 \pm 0.04$ | $-54.04 \pm 0.04$ | $-53.76 \pm 0.02$ |
| $p(z)$, $K = 2K_0$ | $\mathbf{-55.75 \pm 0.04}$ | $\mathbf{-54.90 \pm 0.03}$ | $\mathbf{-54.00 \pm 0.04}$ | $\mathbf{-53.64 \pm 0.02}$ | $\mathbf{-53.49 \pm 0.03}$ |

Table B.15: Test NLL of ResnetVAE, with RealNVP prior of different flow depth $K$. Even $K = 1$, RealNVP prior improves NLLs by about 1 nat. There is no over-fitting for $K$ up to 50. However, we think over-fitting may happen with sufficiently powerful prior, 50-blocks of RealNVP may simply not be deep enough.

| Flow depth | Datasets | | | |
|---|---|---|---|---|
| | StaticMNIST | MNIST | FashionMNIST | Omniglot |
| 0 | $82.95 \pm 0.09$ | $81.07 \pm 0.03$ | $226.17 \pm 0.05$ | $96.99 \pm 0.04$ |
| 1 | $81.76 \pm 0.04$ | $80.02 \pm 0.02$ | $225.27 \pm 0.03$ | $96.20 \pm 0.06$ |
| 2 | $81.30 \pm 0.02$ | $79.58 \pm 0.02$ | $224.78 \pm 0.02$ | $95.35 \pm 0.06$ |
| 5 | $80.64 \pm 0.06$ | $79.09 \pm 0.02$ | $224.37 \pm 0.01$ | $94.47 \pm 0.01$ |
| 10 | $80.26 \pm 0.05$ | $78.75 \pm 0.01$ | $224.18 \pm 0.01$ | $93.92 \pm 0.02$ |
| 20 | $79.99 \pm 0.02$ | $78.58 \pm 0.01$ | $224.09 \pm 0.01$ | $93.61 \pm 0.04$ |
| 30 | $79.90 \pm 0.05$ | $78.52 \pm 0.01$ | $\mathbf{224.07 \pm 0.01}$ | $93.53 \pm 0.02$ |
| 50 | $\mathbf{79.84 \pm 0.04}$ | $\mathbf{78.49 \pm 0.01}$ | $\mathbf{224.07 \pm 0.01}$ | $\mathbf{93.52 \pm 0.02}$ |

Table B.16: Average test ELBO ("*elbo*"), reconstruction loss ("*recons*"), $\mathbb{E}_{p^\star(\mathbf{x})}D_{\mathrm{KL}}(q_\phi(\mathbf{z}|\mathbf{x})\|p_\lambda(\mathbf{z}))$ ("*kl*") and $\mathbb{E}_{p^\star(\mathbf{x})}D_{\mathrm{KL}}(q_\phi(\mathbf{z}|\mathbf{x})\|p_\theta(\mathbf{z}|\mathbf{x}))$ ("*$kl_{z|x}$*") of various DenseVAE. Flow depth $K = 20$.

| | **standard** | | | |
|---|---|---|---|---|
| **Datasets** | *elbo* | *recons* | *kl* | *$kl_{z\|x}$* |
| StaticMNIST | -94.53 | -65.54 | 29.00 | 5.69 |
| MNIST | -88.19 | -62.04 | **26.16** | 3.71 |
| FashionMNIST | -230.81 | -211.71 | **19.11** | 2.22 |
| Omniglot | -113.59 | -78.25 | 35.33 | 7.17 |
| | **RealNVP** $q(z\|x)$ | | | |
| **Datasets** | *elbo* | *recons* | *kl* | *$kl_{z\|x}$* |
| StaticMNIST | -90.78 | **-62.55** | 28.23 | 4.71 |
| MNIST | -85.15 | -58.83 | 26.31 | 2.62 |
| FashionMNIST | -229.53 | -210.35 | 19.18 | 1.75 |
| Omniglot | -108.54 | -73.71 | **34.83** | **5.57** |
| | **RealNVP** $p(z)$ | | | |
| **Datasets** | *elbo* | *recons* | *kl* | *$kl_{z\|x}$* |
| StaticMNIST | **-88.87** | -63.35 | **25.52** | **4.00** |
| MNIST | **-82.57** | **-55.99** | 26.58 | **2.14** |
| FashionMNIST | **-227.72** | **-208.13** | 19.59 | **1.61** |
| Omniglot | **-107.99** | **-73.10** | 34.89 | 5.80 |

Table B.17: Average test ELBO ("*elbo*"), reconstruction loss ("*recons*"), $\mathbb{E}_{p^\star(\mathbf{x})}D_{\mathrm{KL}}(q_\phi(\mathbf{z}|\mathbf{x})\|p_\lambda(\mathbf{z}))$ ("*kl*") and $\mathbb{E}_{p^\star(\mathbf{x})}D_{\mathrm{KL}}(q_\phi(\mathbf{z}|\mathbf{x})\|p_\theta(\mathbf{z}|\mathbf{x}))$ ("*$kl_{z|x}$*") of various ConvVAE. Flow depth $K = 20$.

| | **standard** | | | |
|---|---|---|---|---|
| **Datasets** | *elbo* | *recons* | *kl* | *$kl_{z\|x}$* |
| StaticMNIST | -88.15 | -60.34 | 27.81 | 4.51 |
| MNIST | -85.89 | -58.97 | 26.92 | 3.75 |
| FashionMNIST | -230.43 | -210.24 | **20.18** | 2.92 |
| Omniglot | -104.70 | -66.00 | 38.70 | 6.83 |
| | **RealNVP** $q(z\|x)$ | | | |
| **Datasets** | *elbo* | *recons* | *kl* | *$kl_{z\|x}$* |
| StaticMNIST | -83.92 | -56.67 | **27.25** | **2.82** |
| MNIST | -82.46 | -56.08 | **26.38** | 2.37 |
| FashionMNIST | -228.20 | -207.33 | 20.88 | 2.18 |
| Omniglot | -99.92 | -62.48 | **37.44** | 5.02 |
| | **RealNVP** $p(z)$ | | | |
| **Datasets** | *elbo* | *recons* | *kl* | *$kl_{z\|x}$* |
| StaticMNIST | **-82.91** | **-54.01** | 28.90 | 2.86 |
| MNIST | **-80.42** | **-53.33** | 27.09 | **1.75** |
| FashionMNIST | **-226.65** | **-204.93** | 21.72 | **2.00** |
| Omniglot | **-98.59** | **-59.07** | 39.51 | **4.91** |

Table B.18: Average test ELBO ("*elbo*"), reconstruction loss ("*recons*"), $\mathbb{E}_{p^\star(\mathbf{x})} D_{\mathrm{KL}}(q_\phi(\mathbf{z}|\mathbf{x})\|p_\lambda(\mathbf{z}))$ ("*kl*") and $\mathbb{E}_{p^\star(\mathbf{x})} D_{\mathrm{KL}}(q_\phi(\mathbf{z}|\mathbf{x})\|p_\theta(\mathbf{z}|\mathbf{x}))$ ("*$kl_{z|x}$*") of various ResnetVAE. Flow depth $K = 20$.

| | **standard** | | | |
|---|---|---|---|---|
| **Datasets** | *elbo* | *recons* | *kl* | *$kl_{z|x}$* |
| StaticMNIST | -87.61 | -60.09 | 27.52 | 4.67 |
| MNIST | -84.62 | -58.70 | 25.92 | 3.55 |
| FashionMNIST | -228.91 | -208.94 | **19.96** | 2.74 |
| Omniglot | -104.87 | -66.98 | 37.89 | 7.88 |
| | **RealNVP** $q(z|x)$ | | | |
| **Datasets** | *elbo* | *recons* | *kl* | *$kl_{z|x}$* |
| StaticMNIST | -84.72 | -58.10 | **26.63** | 3.75 |
| MNIST | -81.95 | -56.15 | **25.80** | 2.42 |
| FashionMNIST | -227.16 | -206.61 | 20.54 | 2.14 |
| Omniglot | -100.30 | -63.34 | **36.96** | 6.00 |
| | **RealNVP** $p(z)$ | | | |
| **Datasets** | *elbo* | *recons* | *kl* | *$kl_{z|x}$* |
| StaticMNIST | **-82.85** | **-54.32** | 28.54 | **2.87** |
| MNIST | **-80.34** | **-53.64** | 26.70 | **1.76** |
| FashionMNIST | **-225.97** | **-204.66** | 21.31 | **1.88** |
| Omniglot | **-99.60** | **-61.21** | 38.39 | **5.99** |

Table B.19: Average test ELBO ("*elbo*"), reconstruction loss ("*recons*"), $\mathbb{E}_{p^\star(\mathbf{x})} D_{\mathrm{KL}}(q_\phi(\mathbf{z}|\mathbf{x})\|p_\lambda(\mathbf{z}))$ ("*kl*") and $\mathbb{E}_{p^\star(\mathbf{x})} D_{\mathrm{KL}}(q_\phi(\mathbf{z}|\mathbf{x})\|p_\theta(\mathbf{z}|\mathbf{x}))$ ("*$kl_{z|x}$*") of various PixelVAE. Flow depth $K = 20$.

| | **standard** | | | |
|---|---|---|---|---|
| **Datasets** | *elbo* | *recons* | *kl* | *$kl_{z|x}$* |
| StaticMNIST | -81.06 | -69.02 | **12.03** | 1.59 |
| MNIST | -79.88 | -68.73 | **11.15** | 1.24 |
| FashionMNIST | -225.60 | -214.15 | **11.45** | 1.38 |
| Omniglot | -91.58 | -82.80 | 8.78 | 1.75 |
| | **RealNVP** $q(z|x)$ | | | |
| **Datasets** | *elbo* | *recons* | *kl* | *$kl_{z|x}$* |
| StaticMNIST | -80.65 | -67.91 | 12.75 | **1.57** |
| MNIST | -79.59 | -68.17 | 11.42 | 1.18 |
| FashionMNIST | -225.05 | -213.41 | 11.64 | **1.24** |
| Omniglot | **-91.26** | -83.17 | **8.10** | 1.57 |
| | **RealNVP** $p(z)$ | | | |
| **Datasets** | *elbo* | *recons* | *kl* | *$kl_{z|x}$* |
| StaticMNIST | **-80.60** | **-62.22** | 18.38 | 1.68 |
| MNIST | **-79.28** | **-64.41** | 14.87 | **1.13** |
| FashionMNIST | **-224.65** | **-210.16** | 14.49 | 1.26 |
| Omniglot | -91.78 | **-79.70** | 12.07 | 2.16 |

Table B.20: Test NLL of ResnetVAE, with prior trained by: *joint* training, *iterative* training, *post-hoc* training, and standard VAE ("none") as reference. Flow depth $K = 20$.

| | ResnetVAE | | | |
|---|---|---|---|---|
| **Datasets** | joint | iterative | post-hoc | none |
| StaticMNIST | **79.99 ± 0.02** | 80.63 ± 0.02 | 80.86 ± 0.04 | 82.95 ± 0.09 |
| MNIST | **78.58 ± 0.01** | 79.61 ± 0.01 | 79.90 ± 0.04 | 81.07 ± 0.03 |
| FashionMNIST | **224.09 ± 0.01** | 224.88 ± 0.02 | 225.22 ± 0.01 | 226.17 ± 0.05 |
| Omniglot | **93.61 ± 0.04** | 94.43 ± 0.11 | 94.87 ± 0.05 | 96.99 ± 0.04 |

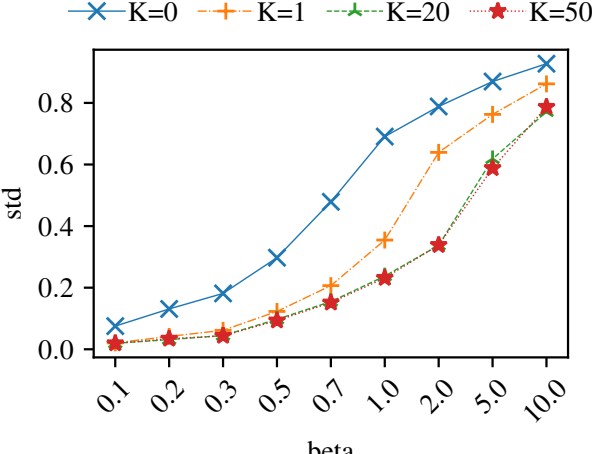

Figure B.6: Avg. *std* of $q_\phi(\mathbf{z}|\mathbf{x})$ of $\beta$-ResnetVAE trained with different $\beta$ and prior flow depth $K$.

## B.13 ADDITIONAL RESULTS: TWO LEVEL HIERARCHICAL VAE WITH FLOW PRIOR

In addition to the comprehensive experiments based on VAE, we also conduct experiments using two-level ResnetHVAE (Tomczak & Welling, 2018), just as the two major previous works (Tomczak & Welling, 2018; Bauer & Mnih, 2019).

**Architectures of ResnetHVAE for binarized datasets**  The observed variable $\mathbf{x}$ is a 3-d tensor, just the same as VAEs. The latent variables $\mathbf{z}_1$ and $\mathbf{z}_2$ are vectors with the same number of dimensions, which is 40 on MNIST and StaticMNIST, and 64 on FashionMNIST and Omniglot. We use $\mathrm{Dim}(\mathbf{z})$ to denote this dimensionality.

The overall factorization for HVAE is:

$$p_\theta(\mathbf{x}, \mathbf{z}_2, \mathbf{z}_1) = p_\theta(\mathbf{x}|\mathbf{z}_1, \mathbf{z}_2)\, p_\theta(\mathbf{z}_1|\mathbf{z}_2)\, p_\lambda(\mathbf{z}_2)$$
$$q_\phi(\mathbf{z}_2, \mathbf{z}_1|\mathbf{x}) = q_\phi(\mathbf{z}_1|\mathbf{z}_2, \mathbf{x})\, q_\phi(\mathbf{z}_2|\mathbf{x})$$

where $p_\lambda(\mathbf{z}_2)$ is either a flow prior or a unit Gaussian prior, depending on the context.

The shared feature network $h_\phi(\mathbf{x})$ for $q_\phi(\mathbf{z}_2|\mathbf{x})$ and $q_\phi(\mathbf{z}_1|\mathbf{x})$ is:

$$\begin{aligned}
h_\phi(\mathbf{x}) = \mathbf{x} \to \mathrm{Resnet}[&28 \times 28 \times 32 \to 28 \times 28 \times 32\\
&\to 14 \times 14 \times 64 \to 14 \times 14 \times 64\\
&\to 7 \times 7 \times 64 \to 7 \times 7 \times 16]\\
\to \mathrm{Flatten}&
\end{aligned}$$

The Gaussian posterior $q_\phi(\mathbf{z}_2|\mathbf{x})$ is derived as:

$$q_\phi(\mathbf{z}_2|\mathbf{x}) = \mathcal{N}\left(\boldsymbol{\mu}_\phi(h_\phi(\mathbf{x})), \boldsymbol{\sigma}_\phi^2(h_\phi(\mathbf{x}))\,\mathbf{I}\right)$$
$$\boldsymbol{\mu}_\phi(h_\phi(\mathbf{x})) = h_\phi(\mathbf{x}) \to \mathrm{Linear}[\mathrm{Dim}(\mathbf{z})]$$
$$\log \boldsymbol{\sigma}_\phi(h_\phi(\mathbf{x})) = h_\phi(\mathbf{x}) \to \mathrm{Linear}[\mathrm{Dim}(\mathbf{z})]$$

The Gaussian posterior $q_\phi(\mathbf{z}_1|\mathbf{x}, \mathbf{z}_2)$ is derived as:

$$q_\phi(\mathbf{z}_1|\mathbf{x}, \mathbf{z}_2) = \mathcal{N}(\boldsymbol{\mu}_\phi(g_\phi(\mathbf{x}, \mathbf{z}_2)), \boldsymbol{\sigma}_\phi^2(g_\phi(\mathbf{x}, \mathbf{z}_2))\mathbf{I})$$
$$\boldsymbol{\mu}_\phi(g_\phi(\mathbf{x}, \mathbf{z}_2)) = g_\phi(\mathbf{x}, \mathbf{z}_2) \to \mathrm{Linear}[\mathrm{Dim}(\mathbf{z})]$$
$$\log \boldsymbol{\sigma}_\phi(g_\phi(\mathbf{x}, \mathbf{z}_2)) = g_\phi(\mathbf{x}, \mathbf{z}_2) \to \mathrm{Linear}[\mathrm{Dim}(\mathbf{z})]$$
$$g_\phi(\mathbf{x}, \mathbf{z}_2) = \mathrm{Concat}[t_\phi(\mathbf{x}), u_\phi(\mathbf{z}_2)] \to \mathrm{Dense}[300]$$
$$t_\phi(\mathbf{x}) = h_\phi(\mathbf{x}) \to \mathrm{Dense}[150 \to 150]$$
$$u_\phi(\mathbf{z}_2) = \mathbf{z}_2 \to \mathrm{Dense}[150 \to 150]$$

The Gaussian prior $p_\theta(\mathbf{z}_1|\mathbf{z}_2)$ is derived as:

$$p_\theta(\mathbf{z}_1|\mathbf{z}_2) = \mathcal{N}\left(\boldsymbol{\mu}_\theta(t_\theta(\mathbf{z}_2)), \boldsymbol{\sigma}_\theta^2(t_\theta(\mathbf{z}_2))\,\mathbf{I}\right)$$
$$\boldsymbol{\mu}_\theta(t_\theta(\mathbf{z}_2)) = t_\theta(\mathbf{z}_2) \to \mathrm{Linear}[\mathrm{Dim}(\mathbf{z})]$$
$$\log \boldsymbol{\sigma}_\theta(t_\theta(\mathbf{z}_2)) = t_\theta(\mathbf{z}_2) \to \mathrm{Linear}[\mathrm{Dim}(\mathbf{z})]$$
$$t_\theta(\mathbf{z}_2) = \mathbf{z}_2 \to \mathrm{Dense}[150 \to 150]$$

For binarized images, the Bernoulli conditional distribution $p_\theta(\mathbf{x}|\mathbf{z}_1, \mathbf{z}_2)$ is derived as:

$$p_\theta(\mathbf{x}|\mathbf{z}_1, \mathbf{z}_2) = \mathrm{Bernoulli}[\boldsymbol{\mu}_\theta(g_\theta(\mathbf{z}_1, \mathbf{z}_2))]$$
$$\log \frac{\boldsymbol{\mu}_\theta(g_\theta(\mathbf{z}_1, \mathbf{z}_2))}{1 - \boldsymbol{\mu}_\theta(g_\theta(\mathbf{z}_1, \mathbf{z}_2))} = g_\theta(\mathbf{z}_1, \mathbf{z}_2) \to \mathrm{LinearConv}_{1\times1}[28 \times 28 \times 1]$$
$$g_\theta(\mathbf{z}_1, \mathbf{z}_2) = \mathrm{Concat}[\mathbf{z}_1, \mathbf{z}_2] \to h_\theta(\mathbf{z})$$
$$\begin{aligned}
h_\theta(\mathbf{z}) = \mathrm{Dense}[784] \to \mathrm{UnFlatten}[&7 \times 7 \times 16]\\
\to \mathrm{DeResnet}[&7 \times 7 \times 64 \to 14 \times 14 \times 64\\
&\to 14 \times 14 \times 64 \to 28 \times 28 \times 32\\
&\to 28 \times 28 \times 32]
\end{aligned}$$

Note the shared feature network $h_\phi(\mathbf{x})$ and the decoder network $h_\theta(\mathbf{z})$ have exactly the same architecture as VAE (see Appendix B.2).

Table B.21: Test negative log-likelihood of ResnetHVAE, compared with ResnetVAE. Flow depth $K = 20$ for $p_\lambda(\mathbf{z})$ in ResnetVAE. For ResnetHVAE, $p_\lambda(\mathbf{z}_2)$ is a flow prior with $K = 20$, and $p_\lambda(\mathbf{z}_1|\mathbf{z}_2)$ is a Gaussian prior.

| **Models** | | **Datasets** | | | |
| | | StaticMNIST | MNIST | FashionMNIST | Omniglot |
|---|---|---|---|---|---|
| ResnetVAE | standard | $82.95 \pm 0.09$ | $81.07 \pm 0.03$ | $226.17 \pm 0.05$ | $96.99 \pm 0.04$ |
| | RNVP $p(z)$ | $79.99 \pm 0.02$ | $78.58 \pm 0.01$ | $\mathbf{224.09 \pm 0.01}$ | $93.61 \pm 0.04$ |
| ResnetHVAE | standard | $81.58 \pm 0.00$ | $80.09 \pm 0.03$ | $225.18 \pm 0.02$ | $95.80 \pm 0.03$ |
| | RNVP $p(z_2)$ | $\mathbf{79.95 \pm 0.01}$ | $\mathbf{78.56 \pm 0.01}$ | $224.11 \pm 0.01$ | $\mathbf{93.47 \pm 0.01}$ |

Table B.22: Average number of *active units* of ResnetHVAE of $z_2$ and $z_1$, with unit Gaussian $p(z_2)$ and $p(z_1)$ ("standard"), and RealNVP $p(z_2)$ and Gaussian $p(z_1)$ ("RNVP $p(z_2)$").

| **ResnetHVAE** | | **Datasets** | | | | | | |
| | StaticMNIST | | MNIST | | FashionMNIST | | Omniglot | |
| | $z_1$ | $z_2$ | $z_1$ | $z_2$ | $z_1$ | $z_2$ | $z_1$ | $z_2$ |
|---|---|---|---|---|---|---|---|---|
| standard | **11.3** | 25 | **8.7** | 18.3 | **16.3** | 16.7 | **12.3** | 57.7 |
| RealNVP $p(z_2)$ | 1.3 | **40** | 0 | **40** | 0.3 | **64** | 0 | **64** |

**Experimental results**   Table B.21 shows the test negative log-likelihood (NLL) of ResnetHVAE, with or without RealNVP $p(\mathbf{z}_2)$, compared with ResnetVAE, with or without RealNVP $p(\mathbf{z})$. We can see that, ResnetHVAE without flow prior has significantly better test NLLs than ResnetVAE without flow prior. However, ResnetHVAE with RealNVP $p(\mathbf{z}_2)$ only has roughly the same test NLLs as ResnetVAE with RealNVP $p(\mathbf{z})$.

Further investigation on the active units of ResnetHVAE (Table B.22) reveals the cause of such experimental results: when coupled with RealNVP $p(\mathbf{z}_2)$, the second latent variable $\mathbf{z}_1$ in ResnetHVAE degenerates, such that no dimension is active in the case of MNIST, FashionMNIST and Omniglot, while almost no dimension is active in the case of StaticMNIST. We also added RealNVP to $\mathbf{z}_1$, but the results showed no difference.

However, we find such degeneration may not always happen. We conducted a pilot experiment of ResnetHVAE with RealNVP $p(\mathbf{z}_2)$ and Gaussian $p(\mathbf{z}_1)$ on Cifar-10. The shape of $\mathbf{z}_2$ and $\mathbf{z}_1$ are both $8 \times 8 \times 16$. The number of active units of $\mathbf{z}_2$ is 1024 (*i.e.*, all dimensions are active), while the number of active units of $\mathbf{z}_1$ is 768 (specifically, these 768 dimensions consist 12 channels of $8 \times 8$ feature maps). The test bpd of this experiment is 3.96. This suggests that the zero active units of $\mathbf{z}_1$ in Table B.22 is not caused by the RealNVP prior. The resnet network architecture we used in our experiments (on the binarized datasets) may have limited the model capacity, such that the ResnetVAE with RealNVP prior is already very close to the limitation of test NLLs in such condition, where adding an auxiliary latent variable will not help.

We also report the *reconstruction loss*, the KL divergence $D_{\mathrm{KL}}(q_\phi(\mathbf{z}_2, \mathbf{z}_1|\mathbf{x})\|p_\theta(\mathbf{z}_2, \mathbf{z}_1))$ and $D_{\mathrm{KL}}(q_\phi(\mathbf{z}_2, \mathbf{z}_1|\mathbf{x})\|p_\theta(\mathbf{z}_2, \mathbf{z}_1|\mathbf{x}))$ of ResnetHVAE with or without RealNVP $p(\mathbf{z}_2)$, which have the same trend as ResnetVAE (Table 6).

Table B.23: Average test ELBO ("*elbo*"), reconstruction loss ("*recons*"), $D_{\mathrm{KL}}(q_\phi(\mathbf{z}_2,\mathbf{z}_1|\mathbf{x})\|p_\theta(\mathbf{z}_2,\mathbf{z}_1))$ ("*kl*"), and $D_{\mathrm{KL}}(q_\phi(\mathbf{z}_2,\mathbf{z}_1|\mathbf{x})\|p_\theta(\mathbf{z}_2,\mathbf{z}_1|\mathbf{x}))$ ("*$kl_{z|x}$*") of of ResnetHVAEs with different prior.

| Datasets | standard | | | | RealNVP $p(z_2)$ | | | |
|---|---|---|---|---|---|---|---|---|
| | *elbo* | *recons* | *kl* | *$kl_{z\|x}$* | *elbo* | *recons* | *kl* | *$kl_{z\|x}$* |
| StaticMNIST | -85.67 | -58.88 | **26.78** | 4.08 | **-82.82** | **-54.11** | 28.71 | **2.87** |
| MNIST | -82.98 | -57.19 | **25.79** | 2.90 | **-80.30** | **-53.55** | 26.76 | **1.74** |
| FashionMNIST | -227.56 | -207.46 | **20.10** | 2.38 | **-225.99** | **-204.48** | 21.51 | **1.88** |
| Omniglot | -103.05 | -65.04 | **38.01** | 7.25 | **-99.37** | **-61.00** | 38.37 | **5.90** |

### B.14 ADDITIONAL RESULTS: LEARNED PRIOR ON LOW POSTERIOR SAMPLES

Rosca et al. (2018) observed that some of the samples from unit Gaussian prior can have low $q_\phi(\mathbf{z})$ and low visual quality, and they name these samples the "*low posterior samples*". Unlike this paper, they tried various approaches to match $q_\phi(\mathbf{z})$ to unit Gaussian $p_\lambda(\mathbf{z})$, including adversarial training methods, but still found *low posterior samples* scattering across the whole prior. We shall see that learning the prior can avoid having high $p_\lambda(\mathbf{z})$ on low posterior samples, and thus we suggest adopting learned prior as a cheap solution to this problem.

We shall first introduce the algorithm proposed by Rosca et al. (2018), which can be used to obtain *low posterior samples* (**z** samples which have low likelihoods on $q_\phi(\mathbf{z})$) from a trained VAE. Their algorithm first samples a large number of **z** from the prior $p_\lambda(\mathbf{z})$, then uses Monte Carlo estimator to evaluate $q_\phi(\mathbf{z})$ on these **z** samples, and finally chooses a certain number of **z** with the lowest $q_\phi(\mathbf{z})$ likelihoods as the *low posterior samples*. They also sampled one **x** from $p_\theta(\mathbf{x}|\mathbf{z})$ for each low posterior sample **z**, and plotted the sample means of these **x** and the histograms of ELBO on these **x**. Although we have found their Monte Carlo estimator for $D_{\mathrm{KL}}(q_\phi(\mathbf{z})\|p_\lambda(\mathbf{z}))$ vulnerable (see Appendix B.8), their *low posterior samples* algorithm only ranks $q_\phi(\mathbf{z})$ for each **z**, and the visual results of their algorithm seem plausible. Thus we think this algorithm should be still convincing enough, and we also use it to obtain *low posterior samples*.

To compare the learned prior with unit Gaussian prior on such *low posterior samples*, we first train a ResnetVAE with unit Gaussian prior (denoted as *standard* ResnetVAE), and then add a *post-hoc trained* RealNVP prior upon this original ResnetVAE (denoted as *post-hoc trained* ResnetVAE). We then obtain 10,000 **z** samples from the *standard* ResnetVAE, and choose 100 **z** with the lowest $q_\phi(\mathbf{z})$ among these 10,000 samples, evaluated on *standard* ResnetVAE. Fixing these 100 **z** samples, we plot the histograms of $\log p_\lambda(\mathbf{z})$ *w.r.t.* *standard* ResnetVAE and *post-hoc trained* ResnetVAE. We also obtain one **x** sample from $p_\theta(\mathbf{x}|\mathbf{z})$ for each **z**, and plot the histograms of ELBO for each **x** (*w.r.t.* the two models) and their sample means. See Fig. B.7.

The **x** samples of *post-hoc trained* ResnetVAE (bottom right) are the same as those of *standard* ResnetVAE (top right), since *post-hoc trained* ResnetVAE has exactly the same $p_\theta(\mathbf{x}|\mathbf{z})$ and $q_\phi(\mathbf{z}|\mathbf{x})$ as *standard* ResnetVAE. However, the *post-hoc trained* prior successfully assigns much lower $\log p_\lambda(\mathbf{z})$ (bottom left) than unit Gaussian prior (top left) on the *low posterior samples*, which suggests that a *post-hoc trained* prior can avoid granting high likelihoods to these samples in the latent space. Note *post-hoc trained* ResnetVAE also assigns slightly lower ELBO (bottom middle) than *standard* ResnetVAE (top middle) to **x** samples corresponding to these *low posterior samples*.

To verify whether a learned prior can avoid obtaining *low posterior samples* in the first place, we obtained *low posterior samples* from a ResnetVAE with *jointly trained* prior (denoted as *jointly trained* ResnetVAE), see Fig. B.8. Compared with Fig. B.7, we can see that $\log p_\lambda(\mathbf{z})$ of these *low posterior samples* and ELBO of the corresponding **x** samples are indeed substantially higher than those of *standard* ResnetVAE and *post-hoc trained* ResnetVAE. However, the visual quality is not perfect, indicating there is still room for improvement.

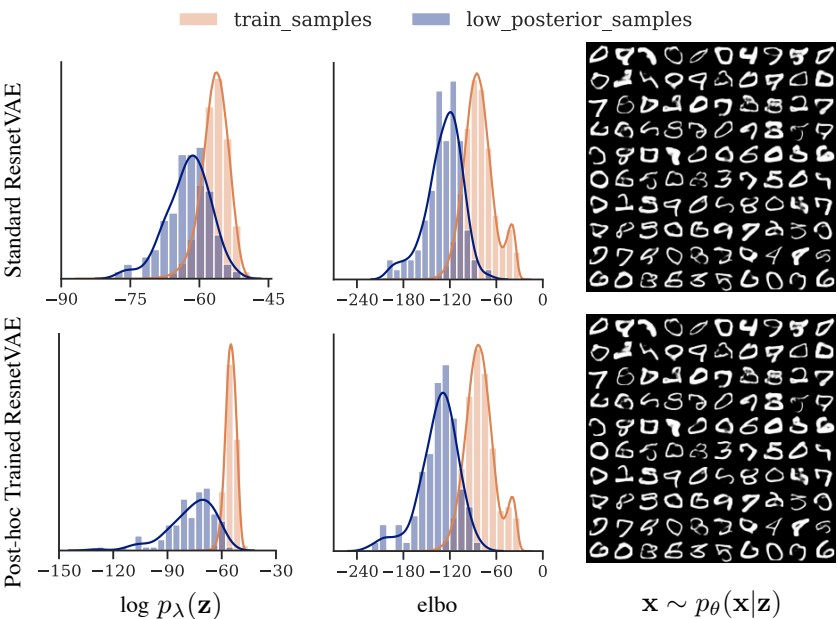

Figure B.7: (left) Histograms of $\log p_\lambda(\mathbf{z})$ of the low posterior samples, (middle) histograms of ELBO of $\mathbf{x}$ samples corresponding to each low posterior sample, and (right) the means of these $\mathbf{x}$, on *standard* ResnetVAE and *post-hoc trained* ResnetVAE.

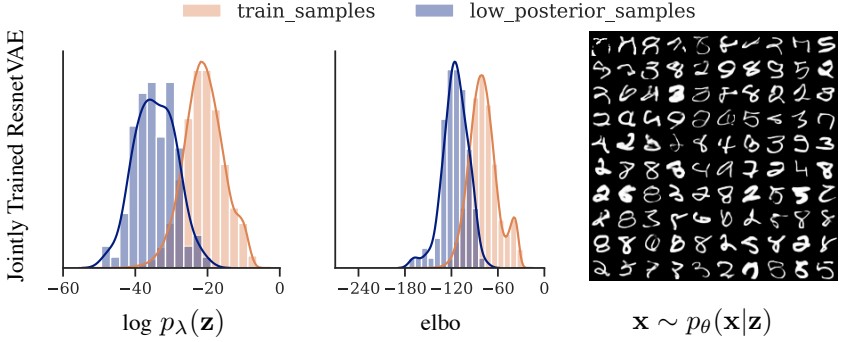

Figure B.8: (left) Histograms of $\log p_\lambda(\mathbf{z})$ of the low posterior samples, (middle) histograms of ELBO of $\mathbf{x}$ samples corresponding to each low posterior sample, and (right) the means of these $\mathbf{x}$, on *jointly trained* ResnetVAE.

## B.15 WALL-CLOCK TIMES

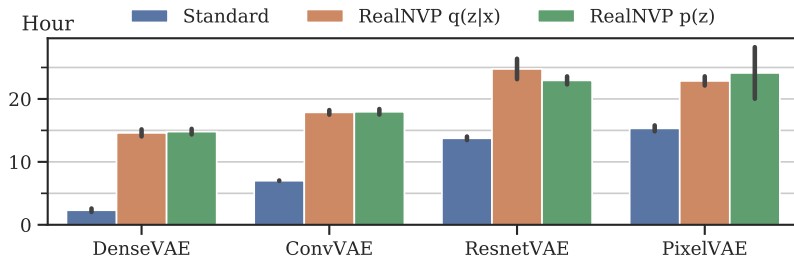

Figure B.9: Average training time of various models on MNIST, flow depth $K = 20$. Black sticks are standard deviation bars. For PixelVAE, training mostly terminates in half way due to early-stopping.

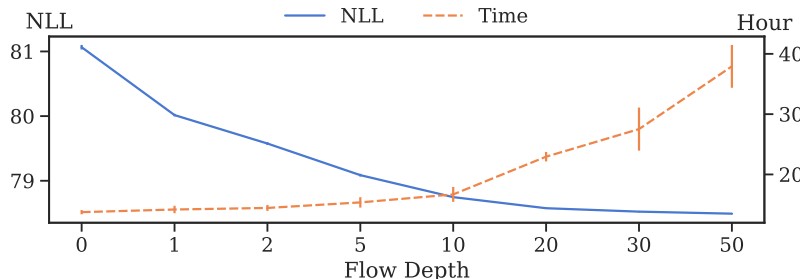

Figure B.10: Average training time and test negative log-likelihood (NLL) of ResnetVAE with RealNVP prior of different flow depth. Vertical sticks are standard deviation bars.

We report the average training time of various models trained on MNIST, see Figs. B.9 and B.10. Each experiment runs on one GTX 1080 Ti graphical card. In Fig. B.9, we can see that the computational cost of RealNVP prior is independent with the architecture of other parts of a model. For complicated architectures like ResnetVAE and PixelVAE, the cost of adding a RealNVP prior is fairly acceptable, since it can bring large improvement. In Fig. B.10, we can see that for $K > 50$, there is likely to be little gain in test NLL, but the computation time will grow even larger. That's why we do not try larger $K$ in our experiments.

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
