# OpenReview forum: "Shallow VAEs with RealNVP Prior Can Perform as Well as Deep Hierarchical VAEs"
_ICLR.cc/2020/Conference — Reject_

### Official Review · AnonReviewer2 · 2019-10-13
**Official Blind Review #2**

**Rating:** 6

**Review:**

This submission shows that using learned autoregressive priors (real NVP) allows shallow VAEs to achieve comparable log-likelihood performances compared to more complex deep VAE architectures.

I found this paper an enjoyable read, and its results quite intriguing. While most of VAE research focuses on building more powerful encoder and decoder architectures, these results show that focusing on a learned prior distribution is as important.

The models introduced in this paper are not novel, but the authors introduce some tricks (gradient and std clipping) that allow them to achieve nearly SOTA results with relatively simple architectures.
I think these tricks should have been demonstrated more in detail in the experiments, for example:
* why is the std clipped exactly at e^-11? What happens if I increase/decrease this number?
* Can the authors clarify the differences between their model and that of Huang et al, which also uses real NVP priors? If I took the exact architecture of Huang et al and used the clipping trick would it perform similarly to your model?
* Would other more complex SOTA models also benefit from your clipping tricks?

A very interesting addition to the paper would be running some experiments on more complex data distributions such as the natural images of celebA or CIFAR10, to understand whether your model could achieve:
(1) similar improvements in terms of  ELBO
(2) more importantly, a quality of the generated samples comparable to deep VAE models such as VAE+IAF or BIVA.

Overall I liked the paper so I am voting towards acceptance. However, while there are a considerable number of experiments in this paper, for me to increase the score I would like to see at least some of the experiments suggested above, since they could help better understand the behavior of VAEs with learned priors and make this an even more impactful paper.


**Experience Assessment:**

I have published in this field for several years.

**Review Assessment: Checking Correctness Of Derivations And Theory:**

I assessed the sensibility of the derivations and theory.

**Review Assessment: Checking Correctness Of Experiments:**

I assessed the sensibility of the experiments.

**Review Assessment: Thoroughness In Paper Reading:**

I read the paper at least twice and used my best judgement in assessing the paper.

---

### Official Review · AnonReviewer3 · 2019-10-24
**Official Blind Review #3**

**Rating:** 3

**Review:**

This paper claims that learning prior from the data could achieve superior performance than using a standard unit Gaussian prior. Experimental results further show that the proposed method could achieve a lower or comparable negative log-likelihood compared to other VAE variants using a complex hierarchical architecture.

I have the following concerns about the paper:

It is widely accepted that the prior serves as the regularization for the Bayesian inference. Using a data-dependent prior is promising to derive a lower NNL than a data-independent prior. However, it would easily lead to an overfitting model with bad generalization, especially for a noisy dataset.

It is claimed in the abstract and conclusion that one latent variable is used for learning the RealNVP prior while the authors use the words "shallow" (refers to few latent variables) in the experiments. So how many latent variables are used in the experiment?

It is listed in the related work that "Huang et al. (2017) applied RealNVP (Dinh et al., 2017) to learn the prior", which means the idea using the learned RealNVP prior is not a new idea. Therefore, what is the contribution of this paper? A detailed discussion is needed to elaborate on the differences from previous methods using a prior learned from the data.

The equation of the aggregated posterior in page 2 after Eq.4 is wrong. The aggregated posterior should be the integration over x instead of z.

The drawn conclusion "using both RealNVP posterior and prior shows no significant advantage over using RealNVP prior only, although the total flow depth of the former variant is twice as large as the latter one" is quite unprofessional. Only one experiment set was conducted with k=20.  One obvious reason is that the current setting with k=20 makes that the model over-parameterized. More comparisons are needed for smaller k.

As claimed in the paper that the clipping would be a navie method to promote overlapping among the posterior. A comparison with a navie baseline using clipping is needed before drawing a conclusion that the learned RealNVP prior is the reason for enhancing the overlapping.

Is the likelihood function p(x|z) a Bernoulli or Gaussian?

"Although BIVA has a much lower NLL on StaticMNIST, in contrast to our paper, the BIVA paper (Maaløe et al., 2019) ...... attributed to having fewer training data". The author should confirm with the authors instead of giving a conjecture in a scientific paper.


**Experience Assessment:**

I have published one or two papers in this area.

**Review Assessment: Checking Correctness Of Derivations And Theory:**

I carefully checked the derivations and theory.

**Review Assessment: Checking Correctness Of Experiments:**

I assessed the sensibility of the experiments.

**Review Assessment: Thoroughness In Paper Reading:**

I read the paper thoroughly.

---

### Official Review · AnonReviewer1 · 2019-10-25
**Official Blind Review #1**

**Rating:** 3

**Review:**

The authors propose to use learned RealNVP with a shallow VAE instead of deep hierarchical VAE, which will not hurt the performance. The authors conduct thorough experiments to backup their claims and hypotheses.

Cons:
1. The writing needs to be improved. The four contributions listed do not have a clear logic flow.
2. The proposed method seems like a combination of previous studies, making the paper more like a technical report.
3. One advantage claimed by the authors is that only one latent variable has clear semantic meanings, which is not explicitly supported by the experiments.

**Experience Assessment:**

I have published one or two papers in this area.

**Review Assessment: Checking Correctness Of Derivations And Theory:**

I assessed the sensibility of the derivations and theory.

**Review Assessment: Checking Correctness Of Experiments:**

I assessed the sensibility of the experiments.

**Review Assessment: Thoroughness In Paper Reading:**

I read the paper at least twice and used my best judgement in assessing the paper.

---

### Decision · Program_Chairs · 2019-12-19

**Decision:**

Reject

**Comment:**

This paper provides an interesting insight into the fitting of variational autoencoders.  While much of the recent literature focuses on training ever more expressive models, the authors demonstrate that learning a flexible prior can provide an equally strong model.  Unfortunately one review is somewhat terse.  Among the other reviews, one reviewer found the paper very interesting and compelling but did not feel comfortable raising their score to "accept" in the discussion phase citing a lack of compelling empirical results in compared to baselines.  Both reviewers were concerned about novelty in light of Huang et al., in which a RealNVP prior is also learned in a VAE.  AnonReviewer3 also felt that the experiments were not thorough enough to back up the claims in the paper.  Unfortunately, for these reasons the recommendation is to reject.  More compelling empirical results with carefully chosen baselines to back up the claims of the paper and comparison to existing literature (Huang et al) would make this paper much stronger.